# Generalized toric polygons, T-branes, and 5d SCFTs

Antoine Bourget[1,2], Andrés Collinucci[3] and Sakura Schäfer-Nameki[4]

**1** Université Paris-Saclay, CNRS, CEA, Institut de physique théorique,
91191, Gif-sur-Yvette, France
**2** Laboratoire de Physique de l'École Normale Supérieure,
PSL University, 24 rue Lhomond, 75005 Paris, France
**3** Service de Physique Théorique et Mathématique,
Université Libre de Bruxelles and International Solvay Institutes,
Campus Plaine C.P. 231, B-1050 Bruxelles, Belgium
**4** Mathematical Institute, University of Oxford,
Woodstock Road, Oxford, OX2 6GG, United Kingdom

## Abstract

5d Superconformal Field Theories (SCFTs) are intrinsically strongly-coupled UV fixed points, whose realization hinges on string theoretic methods: they can be constructed by compactifying M-theory on local Calabi-Yau threefold singularities or alternatively from the world-volume of 5-brane-webs in type IIB string theory. There is a correspondence between 5-brane-webs and toric Calabi-Yau threefolds, however this breaks down when multiple 5-branes are allowed to end on a single 7-brane. In this paper, we extend this connection and provide a geometric realization of brane configurations including 7-branes. A web with 7-branes defines a so-called generalized toric polygon (GTP), which corresponds to combinatorial data that is obtained by removing vertices along external edges of a toric polygon. We identify the geometries associated to GTPs as non-toric deformations of toric Calabi-Yau threefolds and provide a precise, algebraic description of the geometry, when 7-branes are introduced along a single edge. The key ingredients in our analysis are T-branes in a type IIA frame, which includes D6-branes. We show that performing Hanany-Witten moves for the 7-branes on the type IIB side corresponds to switching on semisimple vacuum expectation values on the worldvolume of D6-branes, which in turn uplifts to complex structure deformations of the Calabi-Yau geometries. We test the proposal by computing the crepant resolutions of the deformed geometries, thereby checking consistency with the expected properties of the SCFTs.

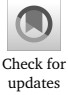

# 1 Introduction and summary

The existence and characterization of interacting superconformal field theories in five space-time dimensions (5d SCFTs) is a remarkable prediction of string theory [1]. Two approaches have emerged that allow the construction of 5d SCFTs within the framework of string theory: the low energy limit of M-theory on $\mathbb{R}^{1,4}$ times a local Calabi-Yau threefold [2], and the world-volume of a brane-web in type IIB string theory [3–12]. The properties of the theory are encoded in the geometry of the threefold in the first case, and in the charges of the external $(p, q)$-5-branes in the second case. A middle ground is the type IIA realization, which involves both geometry and branes [13].

When the CY is toric, there is a precise dictionary between the brane-web and M-theory realization: the charges of the external $(p, q)$-5-branes can be encoded into an integral polygon, which in turn can be seen as the intersection of a toric three-dimensional fan in $\mathbb{R}^3_{x,y,z}$ with the plane $\{z = 1\}$. The toric threefold **X** constructed from this fan is such that the 5d SCFT obtained from M-theory on **X** coincides with that on the world-volume of the brane-web [14]. An example is shown in figure 1.

A systematic geometric exploration and classification of 5d SCFTs was started in [15–38]. These studies reveal many detailed properties of the 5d SCFTs, such as their UV enhanced flavor symmetry, their Coulomb branch (modeled in terms of the crepant resolutions of the Calabi-Yau singularities), but also refined information such as their generalized symmetries [39–44]. What remains somewhat obscure in this framework is the derivation of the full quantum corrected Higgs branch – though some progress in the context of isolated hypersurface Calabi-Yau singularities can be made [32–34, 45, 46].

Not surprisingly, these explorations reveal that only a small class of 5d SCFTs have a realization in terms of toric Calabi-Yau threefolds. If such a toric realization exists, then the geometry of the moduli space of supersymmetric vacua, in particular the Higgs branch (but also Coulomb branch and mixed branches) can be computed exactly – irrespective of whether

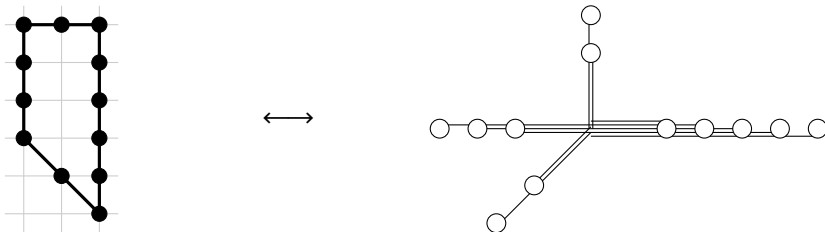

Figure 1: Example of toric polygon (left) and dual brane-web (right) in which lines denote 5-branes and circles denote 7-branes. The 7-branes on which stacks of 5-branes are spaced to emphasize how the 5-brane end, here exactly one 5-brane ends on each 7-brane. This geometry encodes a 5d SCFT of rank 3.

the singularity is isolated or not. The key tool is the connection between the toric geometries and brane-webs, where in the latter these moduli space questions have been determined in [47–53]. In this paper, we report progress in generalizing these methods to a larger class of 5d SCFTs, which have not necessarily a toric description.

In [16], a generalization of toric polygons was introduced: a toric Calabi-Yau threefold can be described in terms of a convex polygon in a square integral lattice embedded into a 2-plane. The polygon associated to a toric geometry has the property that all lattice points along the edges are part of the toric data (corresponding to vertices). We will refer to these as *black dots*. Generalizing this, [16] proposed to also allow some vertices along the edges of the polygon to be unoccupied, which we will refer to as *white dots*. In the dual description, allowing such white dots corresponds in the web to several 5-branes that end on the same 7-brane. For a stack of $n$ 5-branes, the boundary condition is encoded in an integer partition $\lambda$ of $n$. Figure 2 gives an example of a configuration that translates to the [3, 2, 2, 1] partition of 8. This combinatorial data will be referred to as *Generalized Toric Polygons* (GTP), and generalizes the standard toric description. In this framework, the standard toric case considered in the previous paragraph corresponds to a boundary condition where for each charge $(p, q)$, there is an equal number of $(p, q)$-5-branes and $(p, q)$-7-branes, with one 5-brane ending on one 7-brane, i.e. the partition is $\lambda = [1^n]$. This generalization and its implications for characterizing the moduli space of supersymmetric vacua using magnetic quivers were explored in great detail in [32, 45, 53–63]. Note that O5 orientifold planes can also be included in the webs [8, 64–66], but we do not consider this possibility here.

Another way of interpreting GTPs is as non-convex would-be toric polygons. These make sense when certain parameters, which map out the extended Coulomb branch (i.e. gauge couplings and masses of hypers), are turned on, but the non-convexity prevents one from considering the SCFT limit (i.e. from passing to the origin of the extended Coulomb branch). From the dual brane-web point of view, this is resolved using a combination of Hanany-Witten moves and 7-brane monodromies, and this plays a prominent role in the brane-web manipulations of [6, 8, 9, 11, 64, 67–71]. Importantly, not all non-convex polygons can be transformed into GTPs in this way, and it is in general a hard question to decide whether a given polygon can be transformed in this way or not. For this reason, it is simpler to take the GTPs as our starting point.

GTPs can be thought of as generalizations of toric geometries. However, unlike the precise dictionary between the combinatorial data of a toric polygon and the algebraic geometry of the corresponding Calabi-Yau, no such dictionary exists thus far for GTPs. The main purpose of this paper is to develop initial steps in order to close this gap. See figure 3. In particular, in the following, we will determine the algebraic geometric description of GTPs, which have white dots along a single edge.

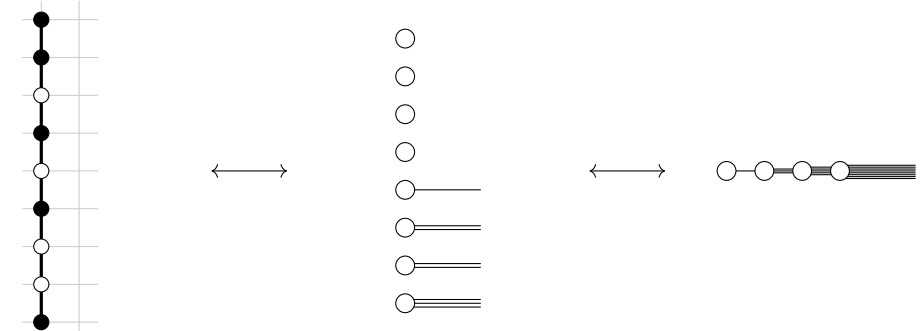

Figure 2: Correspondence between white dots on GTPs (left) and boundary conditions of $(p,q)$ 5-branes on $(p,q)$ 7-branes (middle). Here we have $(p,q) = (1,0)$ and the partition $\lambda = [3,2,2,1]$ of $n = 8$. When we draw brane-webs we usually ignore the detached 7-branes and separate the 7-branes on a stack of 5-branes to show the boundary conditions (right).

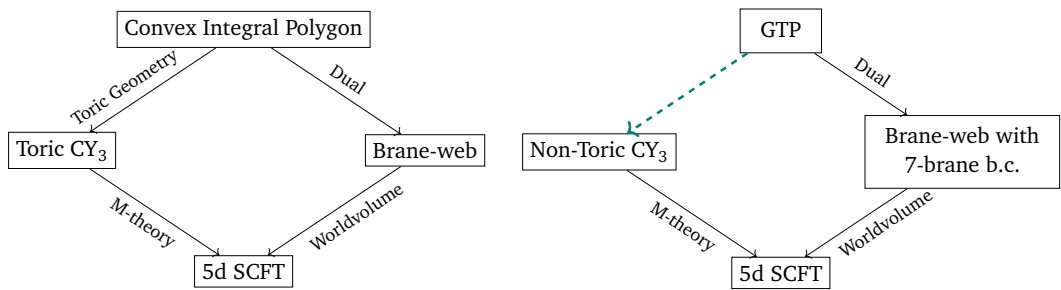

Figure 3: Summary of the main question addressed in this paper. On the left hand side, one starts from a convex polygon with integral vertices. It defines a 5d SCFT in two distinct ways: from M-theory on the associated toric CY, and from the dual brane-web. If on the contrary, as shown on the right hand side, the polygon is not convex, i.e. is a generalized toric polygon (GTP), the toric description is lost. However there still exists a dual brane-web, which has non-trivial boundary conditions on 7-branes, and thus a 5d SCFT. The central goal of this paper is to develop a map (the dashed line in the diagram) from GTPs to (non-toric) geometry.

**Summary.** We now give a schematic summary of the main ideas involved in our proposal. Consider a length $n$ edge of a toric polygon, which we can assume to have vertical orientation. In the brane-web, this corresponds to $n$ parallel semi-infinite D5-branes. In the associated toric threefold, there is an asymptotic region that approaches $\mathbb{C}^2/\mathbb{Z}_n \times \mathbb{C}$. Indeed after a transverse T-duality, the D5-branes become D6-branes, which uplift to $n$-centered Taub-NUT spaces. At strong string coupling $g_s \to \infty$, this becomes $\mathbb{C}^2/\mathbb{Z}_n$. Denoting two longitudinal directions as the $w$-complex plane, we arrive at a local $\mathbb{C}^2/\mathbb{Z}_n \times \mathbb{C}$ patch. M-theory on this geometry gives us $\mathcal{N} = 1$ 7d SYM with SU($n$) gauge group, and we can represent it as the singular hypersurface in $\mathbb{C}^4$ given by

$$uv = z^n, \tag{1}$$

with the $w$-coordinate tagging along. The three adjoint scalars $\phi_{i=1,2,3}$ on the worldvolume of the D6-branes can be grouped into a complex scalar $\Phi = \phi_1 + i\phi_2$, and the remaining real one $\varphi = \phi_3$. In the M-theory uplift, $\Phi$ encodes algebraic deformations to the hypersurface, and $\varphi$ encodes Kähler volumes of resolutions. This grouping is of course arbitrary, and correlates with the arbitrariness of choosing a complex structure on the noncompact K3 in M-theory.

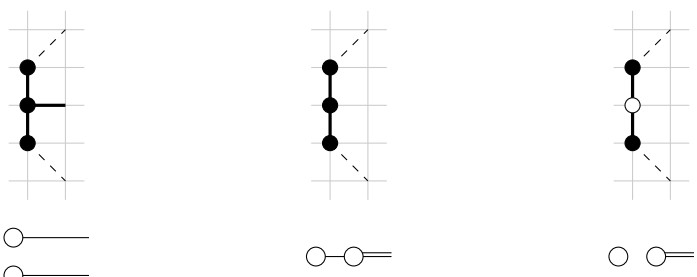

Figure 4: White dot and brane transition for a length 2 edge of a toric polygon.

Having seen this, we can recast the hypersurface as a spectral equation for the complexified adjoint Higgs field

$$uv = \det(\mathbf{1}_n z - \Phi(w)). \tag{2}$$

Switching on constant vevs along the Cartan subalgebra of $\mathfrak{su}(n)$ will deform the equation and unfold the singularity into a deformed K3 times the $w$-plane. However, switching on $w$-dependent vevs will turn this into a *bona fide* noncompact CY threefold. The geometry will be more or less desingularized, depending on the Casimir invariants of $\Phi$ that are switched on. *The claim of this paper, is that white dots correspond to nilpotent elements in $\Phi$.* Note, that switching on a nilpotent $\Phi$ means that the spectral equation remains unchanged. In other words, the D6-branes do not actually move, and the uplifted geometry underlying the M-theory construction remains undeformed. This phenomenon is known as a *T-brane* [72–74]. It is a non-Abelian bound state of branes, whereby the worldvolume gauge group is (partially) Higgsed, but the geometry of the branes is unaltered. However, the physics is of course impacted by this T-brane, as we shall see momentarily.

To give a simple example, take a vertical edge with $n+1$ dots, and replace the second black dot from the top with a white dot as shown in figure 4. In terms of the 5-branes, this corresponds to forming a bound state between the two uppermost branes, and sending a suspended 5-brane segment to infinity. From the D6-brane viewpoint, the bound state is understood as switching on a vev for $\Phi$ along the minimal nilpotent orbit of $\mathfrak{su}(n)$

$$\langle \Phi \rangle = \begin{pmatrix} 0 & 1 & & \\ 0 & 0 & & \\ & & & \ddots \end{pmatrix}. \tag{3}$$

This binds the first two D6-branes, and partially Higgses

$$\mathfrak{su}(n) \rightarrow \mathfrak{s}\left(\mathfrak{u}(1) \oplus \mathfrak{u}(n-2)\right). \tag{4}$$

This is an example of the mechanism known as *brane recombination*: Two stacks of branes, each with their own center of masses, essentially merge into one stack. The fact that the rank of the effective gauge group drops by one signals that two independent centers of mass have become one independent center of mass. More generally, we said above that a distribution of white dots on the edge is encoded in a partition $\lambda$ of $n$. Our claim is that each such partition $\lambda$ of $n$ translates into a vev for $\Phi$ along an element in the nilpotent orbit $\mathcal{O}_\lambda$ of $\mathfrak{su}(n)$ that is uniquely characterized by $\lambda$. The unbroken 7d gauge group on the D6-branes, which will correspond to a subgroup of the total 5d flavor group, is then broken to the commutant of this nilpotent element.

So far, our discussion parallels the picture developed several years ago in [75,76], in the 6d SCFT context, whereby geometric data (about elliptic fibrations) was supplemented by nilpotent orbits, which would partially Higgs an original theory and trigger various RG flows. At



this point, the reader might object that simply claiming that a white dot translates to a nilpotent vev is not very interesting or verifiable, since that data will be invisible to the geometry. While this is true, from the 5-brane-web perspective we know that a white dot opens up the possibility to perform Hanany-Witten type transitions that were not possible in the presence of black dots only. For instance, the following GTPs are related by such a transition where one of the three leftmost 7-branes is moved to the right:

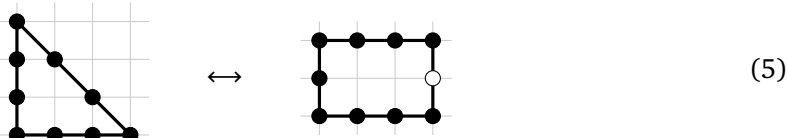

$$\tag{5}$$

Such HW type transitions provide non-trivial tests of our proposal: HW moves correspond to changing the positions of branes, which in turn will impact the dual geometry. We identify the subset of complex structure deformations that are associated to these nilpotent vevs of $\Phi$. They are realized in terms of vevs of $\Phi$ along a slice transverse to the nilpotent vev (inside the full Lie algebra, not the nilpotent cone), known as the *Slodowy slice*. By switching on such a vev, a subset of possible Casimir invariants will become non-zero, leading to a deformation of the geometry, which is given by the spectral equation of the the Higgs field.

For instance, in the simple case of $\mathfrak{u}(2)$, we take the initial nilpotent vev along the minimal orbit

$$\Phi_0 = \begin{pmatrix} 0 & 1 \\ 0 & 0 \end{pmatrix}. \tag{6}$$

The M-theory uplifted geometry corresponds to $\mathbb{C}^2/\mathbb{Z}_2$. The Slodowy slice is given by matrices of the form

$$\Phi = \begin{pmatrix} 0 & 1 \\ a & 0 \end{pmatrix}, \quad \text{with} \quad a \in \mathbb{C}. \tag{7}$$

This is the prototype example of the *brane recombination* mechanism introduced in the previous page: In this case, two independent D-branes have 'merged' into one. Geometrically, they stil occupy the same locus, $z = 0$, however, the gauge algebra has been Higgsed

$$\mathfrak{u}(2) \mapsto \mathfrak{u}(1). \tag{8}$$

So, the initial setup has rank two, corresponding to two independent centers of mass, and the final configuration has rank one, corresponding to the merged object. The characteristic polynomial of this Higgs field is now non-trivial, and the M-theory geometry deforms as follows

$$uv = z^2 \quad \longrightarrow \quad uv = z^2 + a. \tag{9}$$

The present paper elucidates this for all GTPs, which allow for a IIA-description, i.e. whose white dots are along a single edge of the GTP. Dually, all 7-branes are parallel, i.e. mutually local. The toy-model where Slodowy slices appear is generalized in the following way. Higgs branches are symplectic singularities [77], to which one can associate a Hasse diagram of symplectic leaves [78–80]. *In terms of this diagram, the T-brane data select a new, lowest leaf of the foliation, and the transverse slice to that leaf is the total space of a fibration over the complex structure moduli space of the deformed Calabi-Yau threefold.* See for instance figure 5, where the deformations of the $T_4$ 5d SCFT, realized on the threefold $W_1 W_2 W_3 = Z^4$, are displayed, along with the effect on the Higgs branches.

The results of this paper could in principle be phrased purely in the language of stacks of D5 branes ending on D7 branes, dualized to a D6 setup. However, the GTP formalism is adopted as it paves the way towards the necessary generalizations to encompass arbitrary 5d SCFTs. In

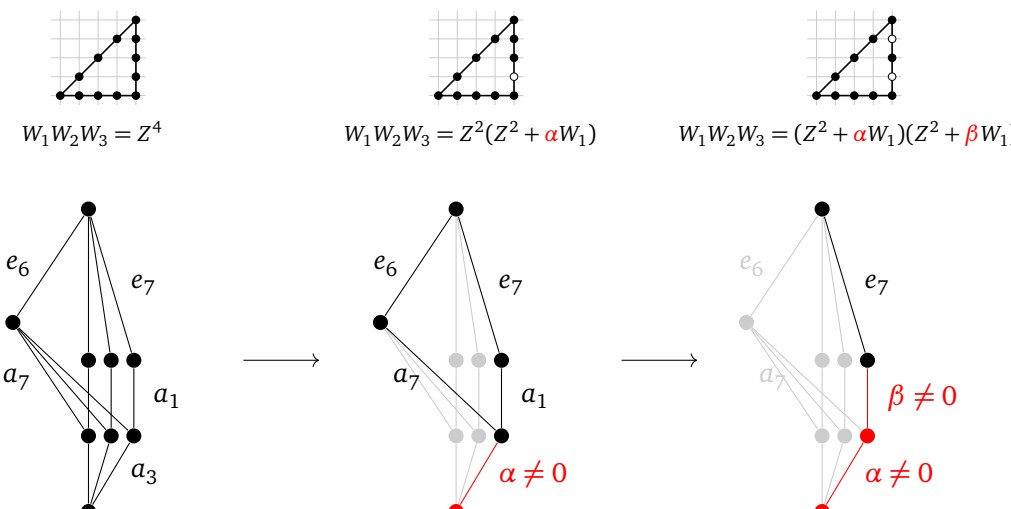

Figure 5: Three GTPs are shown on the first line, and below the algebraic equations characterizing the associated Calabi-Yau threefold geometry. The model on the left is a toric threefold. The other two, non-toric GTPs, are characterized in terms of deformations. Each of these geometries defines a 5d SCFT. The Hasse diagrams of symplectic singularities for the Higgs branch of these 5d SCFTs are shown below. The vertices represent symplectic leaves. For transverse slices we use a standard notation where the closure of the minimal nilpotent orbit of a simple Lie algebra is denoted using the lowercase form of the name of the algebra, e.g. $e_7$ for algebra $E_7$. In red are drawn the effects of the deformations.

future work we will aim to generalize this to arbitrary GTPs, with mutually non-local 7-branes. We conjecture that the above picture of transverse slices in the Higgs branch extends to this situation. Eventually we hope to develop a succinct description of the algebraic geometry of GTPs, as they exist for toric polygons: A precise map between the combinatorial data and the basic algebraic geometry, such as the set of divisors, curves, intersection numbers.

**Plan.** In the rest of the paper, we spell out the details of the construction. An essential tool is the T-brane, which is reviewed in section 2. The bulk of the construction is then carried out explicitly in a representative example – that of the $T_n$ SCFTs – in section 3, before generalizing to any GTP with white dots on a single edge in section 4. As a first check, we reproduce there the transition (5). Finally, in section 5 we provide consistency checks, by computing the resolutions of the deformed threefold geometries. This shows agreement of the UV flavor symmetry of the SCFT with the one expected from the brane-web (and resulting Higgs branch).

## 2 T-branes and Kraft-Procesi transitions

### 2.1 T-brane basics

Consider $n$ parallel D7-branes in type IIB string theory on flat space. The transverse space is the complex plane with coordinate $z$, which has coordinate ring $R = \mathbb{C}[z]$. We call $z_1, \ldots, z_n$ the positions of the $n$ branes. The stack of branes can be described as a D9/$\overline{\text{D9}}$-brane tachyon condensate, which is defined mathematically as the cokernel of the tachyon map

$$R^{\oplus n} \xrightarrow{\ T\ } R^{\oplus n}, \tag{10}$$

where $T = \text{Diag}(z - z_1, \ldots, z - z_n)$. This means that the D7-branes correspond to the sheaf[1] $\mathcal{S}$ in the short exact sequence

$$0 \longrightarrow R^{\oplus n} \xrightarrow{\ T\ } R^{\oplus n} \longrightarrow \mathcal{S} \longrightarrow 0. \tag{11}$$

The matter on the system of D7-branes is described by fluctuations of the tachyon, $\delta T$, which are defined up to linearized gauge transformations. This corresponds to a self-Ext[1] computation for the complex (10), i.e. morphisms between the complex and the shifted version of that same complex,

$$
\begin{array}{ccc}
& R^{\oplus n} \xrightarrow{\ T\ } R^{\oplus n} & \\
{\scriptstyle \alpha_L} \nearrow & \downarrow {\scriptstyle \delta T} \quad \swarrow {\scriptstyle \alpha_R} & \\
R^{\oplus n} \xrightarrow{\ T\ } R^{\oplus n} &
\end{array}
\tag{12}
$$

up to homotopies,

$$\delta T \sim \delta T - T \cdot \alpha_L + \alpha_R \cdot T. \tag{13}$$

Concretely, this means $\delta T$ is valued in the quotient ring of $n \times n$ matrices $\text{Mat}_n(R)$ modulo the two matrix ideals in $R$ generated by left and right multiplication by $T$,

$$\delta T \in \text{Mat}_n(R)/(T\cdot, \cdot T). \tag{14}$$

Note also that the tachyon map $T$ can be expressed as a matrix given a choice of basis for the D9 gauge bundle and the $\overline{\text{D9}}$ gauge bundle. These choices are independent, which means algebraically that only the equivalence class of $T$ under the equivalence relation

$$T \sim G_L \cdot T \cdot G_R^{-1}, \qquad G_L, G_R \in \text{GL}(n, \mathbb{C}[z]), \tag{15}$$

matters.

A canonical representative of each such equivalence class is given by the Smith Normal Form (SNF) computed in the ring $R = \mathbb{C}[z]$, i.e. any matrix $T$ with entries in polynomials in $z$ is equivalent under (15) to a unique diagonal matrix

$$\text{SNF}(T) := \text{diag}(p_1, \ldots, p_r, 0, \ldots, 0), \tag{16}$$

where the $p_i$ are monic polynomials[2] in $z$ such that $p_i$ divides $p_{i+1}$ for all $i$. This gives an algorithmic way to decide whether two tachyon maps encode equivalent physical systems. Several examples are given below.

**Example.** To illustrate the discussion of the previous paragraph, consider the case $n = 2$. The tachyon matrix is

$$T = \begin{pmatrix} z - z_1 & 0 \\ 0 & z - z_2 \end{pmatrix}, \tag{17}$$

and the fluctuations belong to

$$
\delta T \in \begin{pmatrix} \frac{R}{(z-z_1)} & \frac{R}{(z-z_1, z-z_2)} \\ \frac{R}{(z-z_1, z-z_2)} & \frac{R}{(z-z_2)} \end{pmatrix} \simeq
\begin{cases}
\begin{pmatrix} \mathbb{C} & 0 \\ 0 & \mathbb{C} \end{pmatrix}, & z_1 \neq z_2, \\[12pt]
\begin{pmatrix} \mathbb{C} & \mathbb{C} \\ \mathbb{C} & \mathbb{C} \end{pmatrix}, & z_1 = z_2.
\end{cases}
\tag{18}
$$

---

[1]We will use the formulation of branes modulo tachyon condensation in terms of the derived category of coherent sheaves throughout this paper. Some introduction to this topic can be found in [81–83].

[2]Unicity is guaranteed up to multiplication by units in the ring; demanding that the polynomials be monic, i.e. have coefficient 1 for the term of highest degree, fixes this redundancy.

This follows from an explicit computation of the quotient (14). For any value of $z_1, z_2$ there is U(1) adjoint matter on each D7-brane, and for $z_1 = z_2$ the U(1)$^2$ gauge symmetry enhances to U(2), and one can have fluctuations in the adjoint of U(2). From the U(1)$^2$ perspective this is simply bifundamental matter. Consider the case $z_1 = z_2 = 0$. We can then activate an off-diagonal term:[3]

$$T_{[1,1]}(z) = \begin{pmatrix} z & 0 \\ 0 & z \end{pmatrix} \quad \mapsto \quad T_{[2]}(z) = \begin{pmatrix} z & 1 \\ 0 & z \end{pmatrix}. \tag{19}$$

After this activation, the fluctuations are reduced to

$$\delta T_{[2]}(z) \in \begin{pmatrix} 0 & 0 \\ 1 & 0 \end{pmatrix} \mathbb{C} \oplus \begin{pmatrix} 1 & 0 \\ 0 & 1 \end{pmatrix} \mathbb{C}. \tag{20}$$

Instead of computing the quotient (14), an alternative is to use the SNF, which reveals the same structure in a slightly different guise. Indeed

$$\mathrm{SNF}\begin{pmatrix} z - z_1 & 0 \\ 0 & z - z_2 \end{pmatrix} = \begin{cases} \begin{pmatrix} 1 & 0 \\ 0 & (z - z_1)(z - z_2) \end{pmatrix}, & z_1 \neq z_2, \\ \begin{pmatrix} z - z_1 & 0 \\ 0 & z - z_1 \end{pmatrix}, & z_1 = z_2, \end{cases} \tag{21}$$

so the fluctuations are valued in

$$\delta T \in \begin{cases} \begin{pmatrix} 0 & 0 \\ 0 & \mathbb{C} \oplus z\mathbb{C} \end{pmatrix}, & z_1 \neq z_2. \\ \begin{pmatrix} \mathbb{C} & \mathbb{C} \\ \mathbb{C} & \mathbb{C} \end{pmatrix}, & z_1 = z_2. \end{cases} \tag{22}$$

The counting of the number of degrees of freedom agrees with (18). Similarly, the effect of activating the off-diagonal term is

$$\mathrm{SNF}\begin{pmatrix} z & 1 \\ 0 & z \end{pmatrix} = \begin{pmatrix} 1 & 0 \\ 0 & z^2 \end{pmatrix}. \tag{23}$$

The only non-trivial coefficient, $z^2$, shows that there are two degrees of freedom for the fluctuation, recovering (20). For larger matrices, this is an efficient way of computing the fluctuations. Note also that (23) shows the appearance of an infrared trivial complex

$$R \xrightarrow{\ 1\ } R \cong 0, \tag{24}$$

and a so-called 'thick brane'

$$R \xrightarrow{\ z^2\ } R. \tag{25}$$

**Multivariable polynomials.** The ring of polynomials in one variable $\mathbb{C}[z]$ has the property that every ideal is principal. In particular, it is a *Bézout ring*, which means by definition that any ideal generated by finitely many generators is principal. The ring $\mathbb{C}[x_1, \ldots, x_n]$ for $n > 1$ on the other hand is not a Bézout ring: it has non-principal finitely generated ideals (for example, the ideal generated by $x_1$ and $x_2$). It turns out the SNF is best defined in Bézout rings. By [84, Theorem 2.1] an SNF does not exist for matrices with coefficients in $\mathbb{C}[x_1, \ldots, x_n]$. Thus, at face value it seems not possible to use the SNF to describe intersecting branes.

---

[3]We use the following notations: given a partition $\lambda = [\lambda_1, \ldots, \lambda_r]$ of an integer $n = \lambda_1 + \ldots + \lambda_r$, we construct the matrix $T_\lambda(z) = z\mathbf{1}_n + J_\lambda$ where $J_\lambda$ is the nilpotent matrix with Jordan blocks of sizes $\lambda_i$.

However, this is not a weakness but a feature, as we now demonstrate. Consider the case of two variables, $x$ and $z$. Physically, this means we are considering branes that share an $\mathbb{R}^{1,5}$ and wrap complex curves in the $(x,z)$-plane. Consider a stack of $n$ branes at $x = 0$ and a stack of $m$ branes at $z = 0$. This is described by the diagonal matrix $\mathrm{diag}(x,\ldots,x,z,\ldots,z)$. We can activate non-diagonal terms, that we call $Q$ and $\tilde{Q}$ as they correspond to strings that yield hypermultiplets at low energy

$$T = \begin{pmatrix} x\mathbf{1}_n & \tilde{Q} \\ Q & z\mathbf{1}_m \end{pmatrix}. \tag{26}$$

In order to pick a canonical diagonal form for this matrix, we need to *make a choice of main variable*. Let us pick $z$. This means we extend the non-Bézout ring $\mathbb{C}[x,z]$ to the ring $\mathbb{C}(x)[z]$, which is Bézout as it is a polynomial ring in *one variable* over the field $\mathbb{C}(x)$. This is simply telling us that poles in $x$ have to be included. We now describe the SNF over this ring. Assume first that the eigenvalues of $Q\tilde{Q}$ are all distinct, call them $\lambda_1,\ldots,\lambda_m$. Then the SNF is

$$\mathrm{diag}\left(x,\ldots,x,1,\ldots,1,\prod_{i=1}^{m}\left(z - \frac{\lambda_i}{x}\right)\right). \tag{27}$$

If some of the eigenvalues coincide, the form of the SNF changes. We can collect this information, which is insensitive to the detailed properties of the eigenvalues, by simply stating that the SNF is

$$\begin{pmatrix} x\mathbf{1}_n & 0 \\ 0 & z\mathbf{1}_n - \frac{Q\tilde{Q}}{x} \end{pmatrix}, \qquad \lambda_i \neq \lambda_j. \tag{28}$$

Thus, from the point of view of the stack of $m$ branes at $z = 0$, the presence of the other stack is felt as a pole for the complex adjoint-valued Higgs field living on the brane at $z = 0$, [85–87]. Note that the situation is symmetric and one could have chosen the other stack as the base one. This is exactly the same arbitrariness we made when writing the ring as a Bézout ring.

## 2.2 Kraft-Procesi transitions

Nilpotent orbits for $\mathfrak{sl}_n$ are in one-to-one correspondence with partitions of $n$, and are partially ordered by inclusion of their closure [88]. The nilpotent orbit associated to a partition $\lambda$ of $n$ is denoted $\mathcal{O}_\lambda$. The partial order corresponds to the well-known *dominance ordering* for partitions,[4] and it can be represented by a Hasse diagram, which indicates the covering relation associated to this partial order. The diagram thus obtained also corresponds to the stratification of the nilpotent cone (the set of all nilpotent matrices) into symplectic leaves [77, 78, 89]. Elementary degenerations between adjacent nilpotent orbits are called *Kraft-Procesi transitions*. In the case of $\mathfrak{sl}_n$ nilpotent orbits, these can be either closures of minimal $\mathfrak{sl}_m$ nilpotent orbits or Kleinian singularities $\mathbb{C}^2/\mathbb{Z}_m$ for $m \leq n$. This can be implemented in brane setups [90, 91], where nilpotent orbit closures are realized as Higgs or Coulomb branches of 3d $\mathcal{N} = 4$ quiver theories.

**Slodowy slices.** Consider the case where $T = z \cdot \mathbf{1}_n + M$ and $M$ is a nilpotent matrix. Equation (13) shows that

$$\delta T \sim \delta T + (\alpha_R - \alpha_L)z + (\alpha_R M - M\alpha_L), \qquad \delta T \in \mathrm{Mat}_n(R). \tag{29}$$

---

[4]The dominance ordering is defined as follows. If $\lambda$ and $\mu$ are two partitions of $n$, we say that $\lambda \geq \mu$ if and only if for all $j$, $\sum_{i=1}^{j} \lambda_j \geq \sum_{i=1}^{j} \mu_j$.

Define $\alpha_+ := \frac{1}{2}(\alpha_R + \alpha_L)$ and $\alpha_- := \frac{1}{2}(\alpha_R - \alpha_L)$ this gives

$$\delta T \sim \delta T + 2\alpha_- z + [\alpha_+, M] + \{\alpha_-, M\}, \qquad \delta T \in \mathrm{Mat}_n(R). \tag{30}$$

The image of the map

$$\begin{aligned} \mathrm{Mat}_n(R) &\to \mathrm{Mat}_n(R), \\ \alpha_- &\mapsto 2\alpha_- z + \{\alpha_-, M\} \end{aligned} \tag{31}$$

contains all of $\mathrm{Mat}_n(zR)$,[5] so we can use $\alpha_-$ to eliminate all $z$-dependence in $\delta T$. We still have the freedom to use $\alpha_+$, which defines an equivalence relation

$$\delta T \sim \delta T + [\alpha_+, M], \qquad \delta T \in \mathrm{Mat}_n(\mathbb{C}). \tag{32}$$

The cokernel of the adjoint action by $M$ has dimension

$$d_\lambda := \sum_i (2i - 1)\lambda_i, \tag{33}$$

where $\lambda$ is the partition that specifies the nilpotent orbit of $M$. If one considers only traceless matrices, $\delta T$ then depends only on $d_\lambda - 1$ parameters.

Note that being in the cokernel of $\mathrm{ad}(M)$ corresponds to commuting with the other nilpotent element in the $\mathfrak{sl}_2$-triple generated by $M$. Thus $M + \delta T$ parameterizes the Slodowy slice $\mathcal{S}_M$ transverse to $M$ (see Appendix A for definitions and a proof of this statement). Note that indeed that

$$\forall M \in \mathcal{O}_\lambda(\mathfrak{sl}_n), \qquad \dim \mathcal{S}_M = d_\lambda - 1. \tag{34}$$

**Kraft-Procesi transitions.** Consider two partitions $\lambda$ and $\mu$, which are immediately adjacent in the partial order – one says that $\lambda$ *covers* $\mu$ if they are adjacent and $\lambda > \mu$. Then $\lambda$ and $\mu$ differ only in two entries, say with indices $i < j$, with $\lambda_i - 1 = \mu_i$ and $\lambda_{i+1} + 1 = \mu_{i+1}$, and one of the two following transitions occurs:

| Condition | Transition name | Transverse slice |
|---|---|---|
| $j = i + 1$ | $A_{\lambda_i - \lambda_j - 1}$ | $\mathbb{C}^2 / \mathbb{Z}_{\lambda_i - \lambda_j}$ |
| $\mu_i = \mu_j$ | $a_{i-j-1}$ | $\overline{\mathcal{O}}_{\min}(\mathfrak{sl}(i-j, \mathbb{C}))$ |

(35)

The first corresponds to a Kleinian singularity, whereas the second is the closure of a minimal nilpotent orbit. The equivalence class of tachyon matrices for a partition $\mu = [\mu_1, \ldots, \mu_r]$ (with $\mu_1 \geq \cdots \geq \mu_r$) is characterized by a common SNF, i.e.

$$\mathrm{SNF}(T_\mu(z)) = \begin{pmatrix} 1 & & & & & & \\ & \ddots & & & & & \\ & & 1 & & & & \\ & & & z^{\mu_r} & & & \\ & & & & \ddots & & \\ & & & & & z^{\mu_1} \end{pmatrix}. \tag{36}$$

Starting from the SNF of partition $\mu$, the Kraft-Procesi transition is realized using

$$\mathrm{SNF}\begin{pmatrix} z^{\mu_j} & \alpha z^{\mu_j - 1} \\ 0 & z^{\mu_i} \end{pmatrix} = \begin{pmatrix} z^{\mu_j - 1} & 0 \\ 0 & z^{\mu_i + 1} \end{pmatrix} = \begin{pmatrix} z^{\lambda_j} & 0 \\ 0 & z^{\lambda_i} \end{pmatrix}, \tag{37}$$

for $\alpha \neq 0$.

Note that this is precisely the tachyon matrix formalism analog of the way Kraft-Procesi transitions are realized in Hanany-Witten brane systems for 3d $\mathcal{N} = 4$ quiver theories in [90].

---

[5]This is easily proved by a recursive argument on the degree.

**Examples.** The equality (23) corresponds to the covering of partition $[1,1]$ by $[2]$, whereby two branes are combined into a thick brane. A less trivial case is the covering of $[2,2]$ by $[3,1]$, where we do not simply have two branes being combined. Rather, one of the two thick branes needs to be broken. This is realized in our framework using (37) as follows:

$$\text{SNF}\begin{pmatrix} z^2 & \alpha z \\ 0 & z^2 \end{pmatrix} = \begin{pmatrix} z^1 & 0 \\ 0 & z^3 \end{pmatrix}, \tag{38}$$

for $\alpha \neq 0$.

When the nilpotent orbit Hasse diagram is linear, one can build matrices that encode all partitions at once. This is the case for $n \leq 5$, where the matrices are given by

$$\begin{pmatrix} z & a_1 \\ 0 & z \end{pmatrix}, \qquad \begin{pmatrix} z & a_1 & 0 \\ 0 & z & a_2 \\ 0 & 0 & z \end{pmatrix}, \qquad \begin{pmatrix} z & a_1 & 0 & 0 \\ 0 & z & za_3 + a_4 & 0 \\ 0 & 0 & z & a_2 \\ 0 & 0 & 0 & z \end{pmatrix}, \tag{39}$$

$$\begin{pmatrix} z & a_1 & 0 & 0 & 0 \\ 0 & z & za_3 & 0 & a_2a_3a_4 \\ 0 & 0 & z & a_2 & 0 \\ -z^3 a_6 & 0 & z^2 a_1 a_3 a_6 & z & -za_4 \\ -za_2a_4a_5(a_1a_2a_3a_6+1) & 0 & a_1^2 a_2^2 a_3^2 a_4 a_5 a_6 & 0 & z \end{pmatrix}. \tag{40}$$

This means that the SNF of a matrix above with $a_1, \ldots, a_r$ non-zero gives precisely the $r$-th partition of $n$, the partitions being totally ordered.

# 3 Example: GTPs for $T_n$ and related models

In this section, we use an intermediate step in correspondence between M-theory on a CY threefold and IIB 5-brane-webs: IIA on a resolved $\mathbb{C}^2/\mathbb{Z}_n$ singularity with D6-branes. This discussion follows the philosophy of [13]. We start in this section with the instructive example of $T_n$, and consider its description as well as GTPs obtained by adding white dots along a single edge.

## 3.1 The setup

Consider the toric local Calabi-Yau defined by the toric diagram with vertices at coordinates $(0,0)$, $(n,0)$ and $(n,n)$, drawn here for $n = 5$:

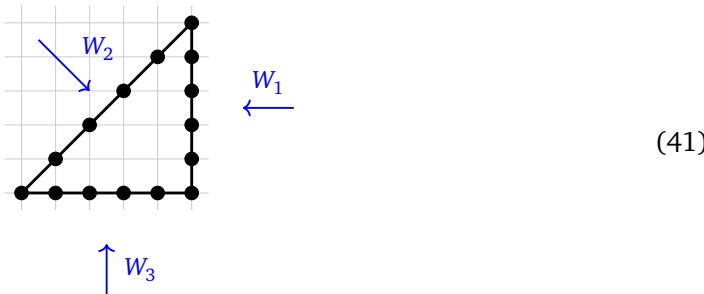

$$\tag{41}$$

The generators of the dual cone are

$$(-1,0,0) \leftrightarrow W_1, \tag{42}$$

$$(1,-1,n) \leftrightarrow W_2, \tag{43}$$

$$(0,1,0) \leftrightarrow W_3, \tag{44}$$

$$(0,0,1) \leftrightarrow Z. \tag{45}$$

The first three generators are vectors normal to the 2-dimensional facets on the fan, drawn in blue arrows when projected on the CY plane in (41). As an algebraic variety, the toric threefold is simply a hypersurface in $\mathbb{C}^4$:

$$W_1 W_2 W_3 = Z^n \quad \subset \quad \mathbb{C}^4\langle W_1, W_2, W_3, Z \rangle. \tag{46}$$

This space has non-isolated singularities, specified by the following intersecting ideals:

$$I_{\text{sing}} = (W_1, W_2, Z) \cap (W_1, W_3, Z) \cap (W_2, W_3, Z). \tag{47}$$

Along each such ideal, there is a family of $A_{n-1}$-singularities. These are in one-to-one correspondence with the three edges of the toric graph.

The singular threefold admits three different (albeit linearly dependent) $\mathbb{C}^*$-actions which act on the coordinates with the following weights:

|  | $W_1$ | $W_2$ | $W_3$ | $Z$ |
|---|---|---|---|---|
| $\mathbb{C}_1^*$ | 0 | 1 | −1 | 0 |
| $\mathbb{C}_2^*$ | 1 | 0 | −1 | 0 |
| $\mathbb{C}_3^*$ | 1 | −1 | 0 | 0 |

$$\tag{48}$$

As explained in toric language in [13], we can define projections $\pi_i$, for $i = 1, 2, 3$, with respect to these actions, and this will bring us down to IIA. Let $(i, j, k)$ be a permutation of $(1, 2, 3)$. The way to reduce along a particular $\mathbb{C}^*$-action $\mathbb{C}_i^*$ is to pick the pair of 'charged' coordinates $W_j$ and $W_k$, and setup a $\mathbb{C}^*$-fibration over an new complex coordinate $V_{jk}$ as follows:

$$\mathbb{C}_i^* : \mathbb{C}[W_1, W_2, W_3, Z] \cong \frac{\mathbb{C}[W_1, W_2, W_3, Z, V_{jk}]}{(W_j W_k - V_{jk})}. \tag{49}$$

Now we can rewrite the threefold in the following presentation:

$$\frac{\mathbb{C}[W_1, W_2, W_3, Z]}{(W_1 W_2 W_3 - Z^n)} \cong \frac{\mathbb{C}[W_1, W_2, W_3, Z, V_{jk}]}{(W_j W_k - V_{jk}; \ W_i V_{jk} - Z^n)}. \tag{50}$$

The IIA reduction is achieved by reducing over the $S^1 \subset \mathbb{C}^*$ action in each case. The non-compact part $\mathbb{R} \subset \mathbb{C}^*$ becomes a transverse direction to the D6-branes. The projection is simply defined as dropping the pair of coordinates $(W_j, W_k)$, leaving us with a local K3 with an $A_{n-1}$ Klein singularity

$$\frac{\mathbb{C}[W_i, V_{jk}, Z]}{(W_i V_{jk} - Z^n)}. \tag{51}$$

To simplify the notations, we switch to the more standard

$$X := W_i, \qquad Y := V_{jk}, \tag{52}$$

so that the $A_{n-1}$ singularity (a local K3) is described by

$$XY = Z^n. \tag{53}$$

There are D6-branes on the locus defined by the ideal

$$I_{\text{D6}} = (Y, Z^n), \tag{54}$$

which we call the D6 ideal. It is, first of all a stack of $n$ non-compact D6-branes. However, since this passes through the singularity, more is at play here, and we need to resolve the local K3 to refine our understanding.

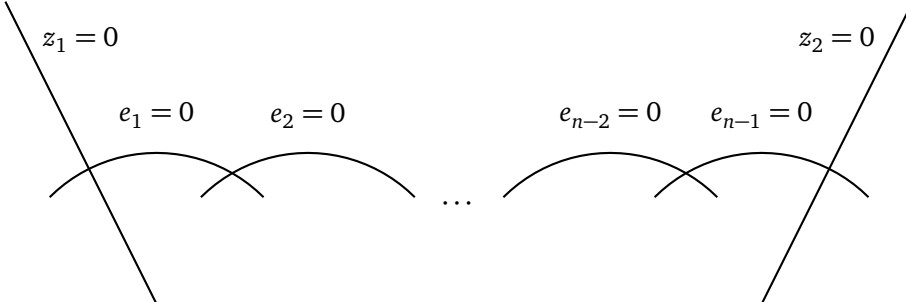

Figure 6: Geometry of the resolved $A_{n-1}$ singularity. Each curved line is a $\mathbb{P}^1$ while the straight lines are the non compact divisors $z_{1,2} = 0$.

In order to describe the resolution of the $A_{n-1}$ orbifold (53), we introduce homogeneous coordinates $(z_1, e_1, \ldots, e_{n-1}, z_2)$ with $n-1$ $\mathbb{C}^*$-actions

$$
\begin{array}{c|cccccccc}
 & z_1 & e_1 & e_2 & e_3 & \cdots & e_{n-3} & e_{n-2} & e_{n-1} & z_2 \\
\hline
\mathbb{C}^*_1 & 1 & -2 & 1 & 0 & \cdots & 0 & 0 & 0 & 0 \\
\mathbb{C}^*_2 & 0 & 1 & -2 & 1 & \cdots & 0 & 0 & 0 & 0 \\
\vdots & & & & & & & & & \\
\mathbb{C}^*_{n-2} & 0 & 0 & 0 & 0 & \cdots & 1 & -2 & 1 & 0 \\
\mathbb{C}^*_{n-1} & 0 & 0 & 0 & 0 & \cdots & 0 & 1 & -2 & 1 \\
\end{array}
\tag{55}
$$

The coordinates are homogeneous with respect to the $n-1$ projective actions, by which the space is quotiented. Each row gives the list of weights of the coordinate with respect to each such action. Just as one must excise particular loci when creating standard projective space, so must one excise a number of loci here.

Specifically, if a coordinate is set to zero, then only one its two 'neighbors' are allowed to vanish. See figure 6. For example, we can have $e_2 = 0$, and $e_3 = 0$, or $e_2 = 0$ and $e_1 = 0$, but the pair $(e_1, e_3)$ does not form a valid ideal for a vanishing locus. In terms of the coordinates $(X, Y, Z)$, we have

$$
X = \prod_{i=0}^{n} e_i^{n-i}, \qquad Y = \prod_{i=0}^{n} e_i^{i}, \qquad Z = \prod_{i=0}^{n} e_i,
\tag{56}
$$

where we have introduced $e_0 := z_1$ and $e_n := z_2$. The locus $e_i = 0$ corresponds to the $i$-th exceptional $\mathbb{P}^1$. The loci $z_1 = 0$ and $z_2 = 0$ correspond to noncompact holomorphic curves intersecting the first and last $\mathbb{P}^1$, respectively. Line bundles over this space are characterized by their first Chern class, which is encoded as $\mathcal{O}(k_1, \ldots, k_{n-1})$. A section of this bundle is a polynomial of homogeneous multi-degree $(k_1, \ldots, k_{n-1})$.

The brane locus (54) is now given by

$$
I_{\mathrm{D6}} = \left( \prod_{i=0}^{n} e_i^{i}, \prod_{i=0}^{n} e_i^{n} \right) = \left( \prod_{i=0}^{n} e_i^{i} \right).
\tag{57}
$$

The interpretation is as follows: there are $i$ D6-brane wrapping the $i$-th $\mathbb{P}^1$, and $n$ non-compact D6-branes on the curve $z_2 = 0$, which intersects the $(n-1)$-th sphere at one point. At the SCFT point, all the $\mathbb{P}^1$'s shrink to zero size. On the Coulomb branch, where the Kähler volumes are non zero, the effective theory can be read from the ideal (57). It is described by the quiver:

$$
\underset{\mathrm{U}(1)}{\circ} \!-\!\!-\!\! \underset{\mathrm{U}(2)}{\circ} \!-\! \cdots \!-\! \underset{\mathrm{U}(n-2)}{\circ} \!-\!\!-\!\! \underset{\mathrm{U}(n-1)}{\circ} \!-\!\!-\!\!-\!\! \underset{n}{\square} \ .
\tag{58}
$$

## 3.2 Tachyon condensation picture

The theory (58) is encoded by via the tachyon condensation/coherent sheaf language as follows. First recall that in terms of complexes, one can describe the branes as follows:

- A brane $B_i$ wrapped on the $i$-th $\mathbb{P}^1$ given by $e_i = 0$ can be described as the cokernel of the complex of line bundles:

$$B_i: \qquad \mathcal{O}(-e_i) \xrightarrow{e_i} \mathcal{O}, \tag{59}$$

  where $\mathcal{O}$ is the structure sheaf over the local K3, and $\mathcal{O}(-e_i)$ is the dual of the line bundle $\mathcal{O}(e_i)$, of which $e_i$ is a section. For instance, $\mathcal{O}(-e_1) = \mathcal{O}(2, -1, 0, \ldots, 0)$.

- A noncompact 'flavor brane' $B_F$ at the locus $z_2 = 0$, intersecting the rightmost $\mathbb{P}^1$ (given by $e_{n-1} = 0$), is given by the following complex:

$$B_n: \qquad \mathcal{O}(-z_2) \xrightarrow{z_2} \mathcal{O}. \tag{60}$$

- More generally, a noncompact D6-brane intersecting the $i$-th exceptional $\mathbb{P}^1$ will be given by the zero-locus of a section of $\mathcal{O}(0, \ldots, 0, 1, 0, \ldots, 0)$, where the '1' is the $i$-th entry.

Define $N = \frac{1}{2}n(n+1)$ D8-branes with gauge bundle $\mathcal{F}_{D8} := \mathcal{O}^{\oplus N}$ and $N$ anti-D8-branes with gauge bundle

$$\mathcal{F}_{\overline{D8}} := \mathcal{O}(2, -1, 0, \ldots, 0) \oplus \mathcal{O}(-1, 2, -1, \ldots, 0)^{\oplus 2} \oplus \cdots \oplus \mathcal{O}(0, \ldots, 0, -1)^{\oplus n}. \tag{61}$$

We then define the tachyon map

$$T: \mathcal{F}_{\overline{D8}} \to \mathcal{F}_{D8}, \tag{62}$$

as a diagonal matrix

$$T = \text{Diag}(e_1 \cdot \mathbf{1}_1, e_2 \cdot \mathbf{1}_2, \ldots, e_{n-1} \cdot \mathbf{1}_{n-1}, z_2 \cdot \mathbf{1}_n). \tag{63}$$

Note that $\det T = Y$, so that the equations of the threefold are

$$W_j W_k = \det T = Y, \qquad XY = Z^n, \tag{64}$$

from which one recover the original equation $W_i W_j W_k = Z^n$. The resulting D6-brane system is defined as the cokernel $\mathcal{S} := \text{cok}(T)$ of this map, which is the locus where $T$ fails to be invertible. So $\mathcal{S}$ is reducible:

$$\mathcal{S} = \bigoplus_{i=1}^{n} \mathcal{O}_{e_i}^{\oplus i}, \tag{65}$$

where $\mathcal{O}_p$ means the structure sheaf with support over $p = 0$. There is an exact sequence

$$\mathcal{F}_{\overline{D8}} \xrightarrow{T} \mathcal{F}_{D8} \longrightarrow \mathcal{S} \longrightarrow 0. \tag{66}$$

**Fluctuations.** The fluctuations around this background are computed as self-extensions, in the same way as in section 2. This means that the fluctuations $\delta T$ of the tachyon $T$ belong to the self-extension group,

$$\delta T \in \text{Ext}^1(\mathcal{S}, \mathcal{S}) = \text{Hom}_{\mathcal{D}(K3)}(\mathcal{S}, \mathcal{S}[1]), \tag{67}$$

where we consider the Homs in the derived category of coherent sheaves on K3 $\mathcal{D}(K3)$, and the $[1]$ means that we shift the complex one step to the left. The matrix $\delta T$ can be decomposed in blocks, and there will be non-zero fluctuations just above and below the diagonal.

In practice, the fluctuations are subjected to three conditions, that we will be using repeatedly in the following sections:

$(i)$ The $\mathbb{C}^*$ weights of the columns of $T$ and $\delta T$ need to be compatible with (63) ;

$(ii)$ The fluctuations are regular sections of the relevant sheaves (no pole on the support locus) ;

$(iii)$ The components of $\delta T$ are subject to the identifications (14).

### 3.3 Example: $T_2$

For concreteness we work out in detail the case $n = 2$ for $T_n$ before treating the general case. The tachyon matrix is

$$\mathcal{O}(2) \oplus \mathcal{O}(-1)^{\oplus 2} \xrightarrow{\;T\;} \mathcal{O}^{\oplus 3} \;, \qquad T = \begin{pmatrix} e_1 & 0 \\ 0 & z_2 \cdot \mathbf{1}_2 \end{pmatrix}, \tag{68}$$

and we give names to the blocks in the fluctuation $\delta T$ in correspondence with the quiver

$$\delta T = \begin{pmatrix} \Phi_1 & \widetilde{Q}_1 \\ Q_1 & \Phi_2 \end{pmatrix}, \qquad \Phi_1 \;\substack{Q_1 \\ \text{1} \;\;\;\;\; \text{2} \\ \widetilde{Q}_1}\; \Phi_2 . \tag{69}$$

The three conditions listed above give

$$\delta T \in \begin{pmatrix} \Gamma(\mathcal{O}(-2)_{(e_1)}) & \Gamma(\mathcal{O}(1)_{(e_1, z_2)}) \cdot \mathbb{C}^{1 \times 2} \\ \Gamma(\mathcal{O}(-2)_{(e_1, z_2)}) \cdot \mathbb{C}^{2 \times 1} & \Gamma(\mathcal{O}(1)_{(z_2)}) \cdot \mathbb{C}^{2 \times 2} \end{pmatrix} = \begin{pmatrix} 0 & \mathbb{C}^{1 \times 2} \cdot z_1 \\ \mathbb{C}^{2 \times 1} \cdot \frac{1}{z_1^2} & \mathbb{C}^{2 \times 2} \cdot z_1 \end{pmatrix}. \tag{70}$$

Here, $\Gamma$ indicates that we take sections of the bundles, and $\mathbb{C}^{n \times m}$ is the set of $n \times m$ matrix of complex numbers. The last equality is easily checked, e.g. for the lower-left entry, the regular sections are generated by rational functions in $z_1$ alone, having $\mathbb{C}^*$-weight $-2$, and poles are allowed as the support is the intersection between two curves where $z_1 \neq 0$. In terms of Ext groups between $B_1$ (see (59)) and $B_2$ (see (60)), we can write (using e.g. a spectral sequence argument)

$$Q_1 \in \text{Ext}^1(B_1, B_2) = H^0(\mathcal{O}(e_1)_{e_1, z_2}) = \left\langle \frac{1}{z_1^2} \right\rangle, \tag{71}$$

$$\widetilde{Q}_1 \in \text{Ext}^1(B_2, B_1) = H^0(\mathcal{O}(z_2)_{e_1, z_2}) = \langle z_1 \rangle. \tag{72}$$

To summarize, the background tachyon plus fluctuation is given by

$$T + \delta T = \begin{pmatrix} e_1 & \widetilde{q}_1 z_1 \\ \frac{q_1}{z_1^2} & z_2 \cdot \mathbf{1}_2 + z_1 \varphi_2 \end{pmatrix}, \tag{73}$$

where we have pulled out all dependencies in $z_1$, $e_1$ and $z_2$, so that $\widetilde{q}_1 \in \mathbb{C}^{1 \times 2}$, $q_1 \in \mathbb{C}^{2 \times 1}$ and $\varphi_2 \in \mathbb{C}^{2 \times 2}$ are pure constants:

$$Q_1 = \frac{q_1}{z_1^2}, \qquad \widetilde{Q}_1 = \widetilde{q}_1 z_1, \qquad \Phi_2 = \varphi_2 z_1. \tag{74}$$

Note that the term $\frac{q_1}{z_1^2}$ ensures that (73) is defined on the locus $z_1 \neq 0$.

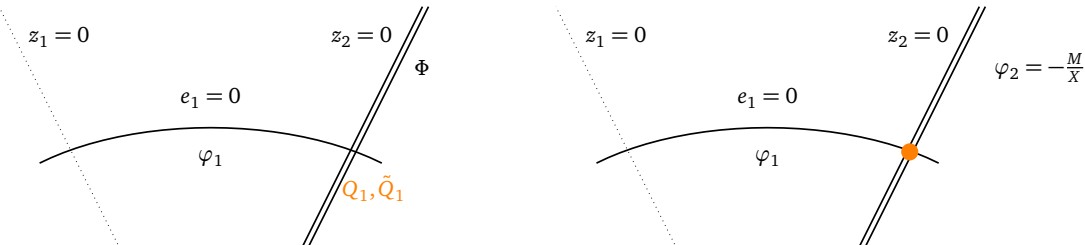

Figure 7: Intersecting branes before and after the transformation that maps the off-diagonal Higgs-field entries $Q_1, \tilde{Q}_1$ to diagonal ones with a pole. The orange dot signals the pole in the Higgs field at $X = 0$.

We can perform basis changes from the left and right, using (15), as follows:

$$\begin{pmatrix} 1 & 0 \\ -\frac{q_1}{z_1^2 e_1} & \mathbf{1}_2 \end{pmatrix} \cdot \begin{pmatrix} e_1 & \tilde{q}_1 z_1 \\ \frac{q_1}{z_1^2} & z_2 \cdot \mathbf{1}_2 \end{pmatrix} \cdot \begin{pmatrix} 1 & -\frac{\tilde{q}_1 z_1}{e_1} \\ 0 & \mathbf{1}_2 \end{pmatrix} = \begin{pmatrix} e_1 & 0 \\ 0 & z_2 \cdot \mathbf{1}_2 - \frac{M}{z_1 e_1} \end{pmatrix}. \tag{75}$$

In the last step we have introduced the meson matrix

$$M := q_1 \tilde{q}_1. \tag{76}$$

In the 5d effective field theory, F-term conditions impose that $M$ be nilpotent. This can also be demonstrated mathematically via the so-called *cone construction* in the derived category of coherent sheaves. See [83] for examples of this mechanism.

This diagonalization shows that giving a vev to the meson field can be subsumed into a shift of vev of the adjoint field $\phi$ on the flavor branes, with a pole. Using the coordinate $X = z_1^2 e_1$ (see (56)) on the $z_2 = 0$ plane, we can write this as

$$T + \delta T \sim \begin{pmatrix} e_1 & 0 \\ 0 & z_2 \cdot \mathbf{1}_2 + z_1 \cdot \varphi_2 \end{pmatrix}, \qquad \text{with} \qquad \varphi_2 = -\frac{M}{X}. \tag{77}$$

The transformation from (73) to (77) means that we can regard in an appropriate regime the intersecting branes in figure 7 as a stack of two branes on $z_2 = 0$ with a complex codimension-one defect on its world-volume at $e_1 = 0$. This is in agreement with the findings of [85, 87], where the authors find that a vev of bifundamental fields at the intersection of two branes can be subsumed into a pole for the adjoint of one of the two branes. In the picture, we trade the description in figure 7 on the left with the right.

Up to a change of basis we can take $M$ in canonical Jordan form. There are two possibilities, corresponding to the two partitions of $n = 2$:

$$M_{[1^2]} = \begin{pmatrix} 0 & 0 \\ 0 & 0 \end{pmatrix}, \qquad \text{and} \qquad M_{[2]} = \begin{pmatrix} 0 & 1 \\ 0 & 0 \end{pmatrix}. \tag{78}$$

Consider the latter case,

$$T_{[2]} := \begin{pmatrix} e_1 & \\ & t_{[2]} \end{pmatrix} := \begin{pmatrix} e_1 & 0 & 0 \\ 0 & z_2 & -\frac{1}{z_1 e_1} \\ 0 & 0 & z_2 \end{pmatrix}. \tag{79}$$

We still have $\det T_{[2]} = e z_2^2 = Y$. So the brane configuration has not changed geometrically. Accordingly, the M-theory uplift is still given by (64). However, the tachyon matrix shows us

that the flavor brane no longer carries an SU(2) group. This is the hallmark of a *T-brane*: A non-abelian bound state of branes that does not realize the gauge group that it would naively have given its geometry. This T-brane effect is the IIA counterpart of the change in boundary conditions in the dual type IIB brane-webs

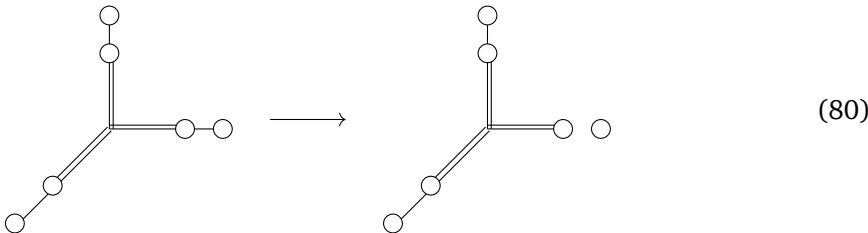

$$\tag{80}$$

In this particular case, once the D5-segment has been sent away, a Hanany-Witten move where the 7-brane detaches completely becomes possible:

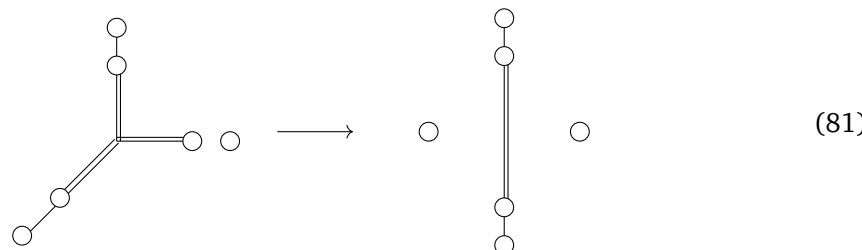

$$\tag{81}$$

How does this translate into the IIA language, i.e. in terms of the tachyon field? Let us see what deformations are available, starting from the new vacuum defined by (79). Using the results of section 2, the fluctuations of $t_{[2]}$ are

$$\delta t_{[2]} = z_1 \cdot \begin{pmatrix} 0 & 0 \\ \alpha & 0 \end{pmatrix}, \qquad \alpha \in \mathbb{C}. \tag{82}$$

Now in order to see how this affects the geometry, we add this perturbation to $T_{[2]}$

$$T_{[2]} + \delta t_{[2]} = \begin{pmatrix} e & 0 & 0 \\ 0 & z_2 & -\frac{1}{z_1 e} \\ 0 & \alpha z_1 & z_2 \end{pmatrix}. \tag{83}$$

Now the geometry (64) is deformed to

$$W_j W_k = \det T = Y + \alpha, \qquad XY = Z^2, \tag{84}$$

which reduces to the hypersurface

$$W_1 W_2 W_3 = Z^2 + \alpha W_i. \tag{85}$$

So the CY threefold is fully desingularized. From the IIA perspective, we see that two flavor branes have recombined with one gauge brane, to give rise to a noncompact brane that can escape the singular locus. This is the same *brane recombination* mechanism described in detail in equations (3)-(8).

This is in full agreement with the 5-brane-web picture, whereby the 7-brane moves off to the left, becomes fully detached from the NS5-branes, and can escape to infinity, see (81). This corresponds precisely to removing the $A_1$ singularity.

To summarize, the white dot is represented in the IIA picture as a nilpotent vev with poles on the flavor D6-stack. The further Hanany-Witten move that actually deforms the M-theory geometry is implemented by switching on a further vev on the flavor stack along the *Slodowy slice* with respect to the initial singular nilpotent vev.

### 3.4 General case: $T_n$

The general $T_n$ case is very similar to the $T_2$ example treated above. The fluctuations around the tachyon background are denoted by fields as follows:

$$
\tag{86}
$$

Generalizing (71) and (72), we find that the 5d hypermultiplets that reside at the intersection of the curves $e_i = 0$ and $e_{i+1} = 0$ are described by

$$
Q_i \in \mathrm{Ext}^1(B_i, B_{i+1}) = H^0(\mathcal{O}(e_i)_{(e_i, e_{i+1})}) = \left\langle \frac{Y}{Z^{i+1}} e_i \right\rangle = \left\langle \prod_{j \neq i} e_j^{j-i-1} \right\rangle, \tag{87}
$$

and

$$
\tilde{Q}_i \in \mathrm{Ext}^1(B_{i+1}, B_i) = H^0(\mathcal{O}(e_{i+1})_{(e_i, e_{i+1})}) = \left\langle \frac{Z^i}{Y} e_{i+1} \right\rangle = \left\langle \prod_{j \neq i+1} e_j^{i-j} \right\rangle. \tag{88}
$$

On the other hand, one can check that $\mathrm{Ext}^1(B_i, B_j) = 0$ for $|i - j| > 1$. Therefore the tachyon matrix $T$, plus the fluctuations $\delta T$, fit schematically (we will provide the explicit form below) into the matrix

$$
T + \delta T = \begin{pmatrix} e_1 \cdot \mathbf{1}_1 & \tilde{Q}_1 & & & & \\ Q_1 & e_2 \cdot \mathbf{1}_2 & \ddots & & & \\ & \ddots & \ddots & \ddots & & \\ & & & e_{n-1} \cdot \mathbf{1}_{n-1} & \tilde{Q}_{n-1} \\ & & & Q_{n-1} & z_2 \cdot \mathbf{1}_n \end{pmatrix}. \tag{89}
$$

As in (74), we introduce the notation

$$
Q_i = q_i \cdot \frac{Y}{Z^{i+1}} e_i, \qquad \tilde{Q}_i = \tilde{q}_i \cdot \frac{Z^i}{Y} e_{i+1}, \tag{90}
$$

such that $q_i$ and $\tilde{q}_i$ are constants. We can also have fluctuations on the diagonal, which we write as

$$
\Phi_i = \varphi_i \cdot \frac{Y}{Z^{i+1}} e_i. \tag{91}
$$

With this notation the F-terms at the $i$th node can be written

$$
\tilde{q}_1 q_1 = 0, \quad \text{and} \quad q_i \tilde{q}_i = \tilde{q}_{i+1} q_{i+1}, \quad i = 1, \ldots, n-2. \tag{92}
$$

We now come back to the reason why (89) is only a schematic form. The $i$-the hyper $(Q_i, \tilde{Q}_i)$ is only well defined at the $(e_i, e_{i+1})$ intersection, but has poles in the nearby patches. Hence, (89) is not well-defined over the whole target space. This is due to the projective nature of the resolved K3. Therefore, we must study it patch by patch. A good local affine coordinate on the locus $e_i = 0$ for the hemisphere where $e_{i+1}$ (respectively $e_{i-1}$) does not vanish is $\frac{Y}{Z^i}$ (respectively $\frac{Z^i}{Y}$):

$$
\tag{93}
$$

$$
e_i = 0 \qquad e_{i+1} = 0
$$

Coordinate $\frac{Y}{Z^i}$        Coordinate $\frac{Z^{i+1}}{Y}$

Note in particular that for $i = N$ (respectively $i = 0$), this is compatible with the affine coordinate on the locus $z_2 = 0$ (resp. $z_1 = 0$) being simply $X$ (resp. $Y$), as chosen in (77).

In the patch that contains the intersection $\{e_i = 0\} \cap \{e_{i+1} = 0\}$, the tachyon fluctuation is expressed as

$$\delta T = \begin{pmatrix} \Phi_i & \widetilde{Q}_i \\ Q_i & \Phi_{i+1} \end{pmatrix} \in \begin{pmatrix} \mathbb{C}^{i \times i} \cdot \frac{Y}{Z^{i+1}} e_i & \mathbb{C}^{i \times (i+1)} \cdot \frac{Z^i}{Y} e_{i+1} \\ \mathbb{C}^{(i+1) \times i} \cdot \frac{Y}{Z^{i+1}} e_i & \mathbb{C}^{(i+1) \times (i+1)} \cdot \frac{Z^i}{Y} e_{i+1} \end{pmatrix}. \tag{94}$$

Using line and row transformations, one finds

$$\begin{pmatrix} \mathbf{1}_i & \widetilde{q}_i \frac{Z^i}{Y} \\ -q_i \frac{Y}{Z^{i+1}} & \mathbf{1}_{i+1} - \frac{q_i \widetilde{q}_i}{Z} \end{pmatrix} \begin{pmatrix} e_i \left( \mathbf{1}_i - \frac{\widetilde{q}_i q_i}{Z} \right) & 0 \\ 0 & e_{i+1} \cdot \mathbf{1}_{i+1} \end{pmatrix} \begin{pmatrix} \mathbf{1}_i & -\widetilde{q}_i \frac{Z^i e_{i+1}}{Y e_i} \\ q_i \frac{Y e_i}{Z^{i+1} e_{i+1}} & \mathbf{1}_{i+1} - \frac{q_i \widetilde{q}_i}{Z} \end{pmatrix}$$
$$= \begin{pmatrix} e_i \cdot \mathbf{1}_i & 0 \\ 0 & e_{i+1} \left( \mathbf{1}_{i+1} - \frac{q_i \widetilde{q}_i}{Z} \right) \end{pmatrix}. \tag{95}$$

Effectively, this transforms a pole of the form

$$\begin{pmatrix} e_i \cdot \mathbf{1}_i + \frac{Y}{Z^{i+1}} e_i \varphi_i & 0 \\ 0 & e_{i+1} \cdot \mathbf{1}_{i+1} \end{pmatrix}, \qquad \varphi_i = \frac{-\widetilde{q}_i q_i}{(Y/Z^i)}, \tag{96}$$

into a pole of the form

$$\begin{pmatrix} e_i \cdot \mathbf{1}_i & 0 \\ 0 & e_{i+1} \mathbf{1}_{i+1} + \frac{Z^i}{Y} e_{i+1} \varphi_{i+1} \end{pmatrix}, \qquad \varphi_{i+1} = \frac{-q_i \widetilde{q}_i}{(Z^{i+1}/Y)}. \tag{97}$$

Let us introduce the $n \times n$ meson matrix

$$M = q_{n-1} \widetilde{q}_{n-1}. \tag{98}$$

As argued for the $T_2$ previously, this meson matrix must be nilpotent. We can characterize the tachyon fluctuation entirely by the last pole (97) for $i = n-1$, which gives

$$\varphi_n = -\frac{M}{X}, \tag{99}$$

consistently with what we found for the $n = 2$ case in (77). In summary, the bifundamental matter between the various branes can be subsumed under a shift of the 7d $SU(n)$-adjoint Higgs $\varphi_n$, as a simple pole with residue equal to a nilpotent element $M$. Since $M^n = 0$, we still have

$$\det T = \prod_{i=1}^{n} e_i^i = Y. \tag{100}$$

Hence, the brane locus remains unscathed despite the activation of the matter fields.

## 3.5 Interpretation in terms of generalized toric polygons

We saw in the last subsection that the geometry of the threefold can be affected by the presence of a pole of the form (99). This pole can be encoded in two ways: in the nilpotent meson matrix $M$, or in the list of hypermultiplet deformations $q_i$ and $\widetilde{q}_i$ satisfying the relations (92). Let us call $\mathcal{Q}$ the set

$$\mathcal{Q} = \{ \mathbf{q} = (q_1, \widetilde{q}_1, \ldots, q_{n-1}, \widetilde{q}_{n-1}) \in \mathbb{C}^{2 \times 1} \times \cdots \times \mathbb{C}^{(n-1) \times n} | (92) \}. \tag{101}$$

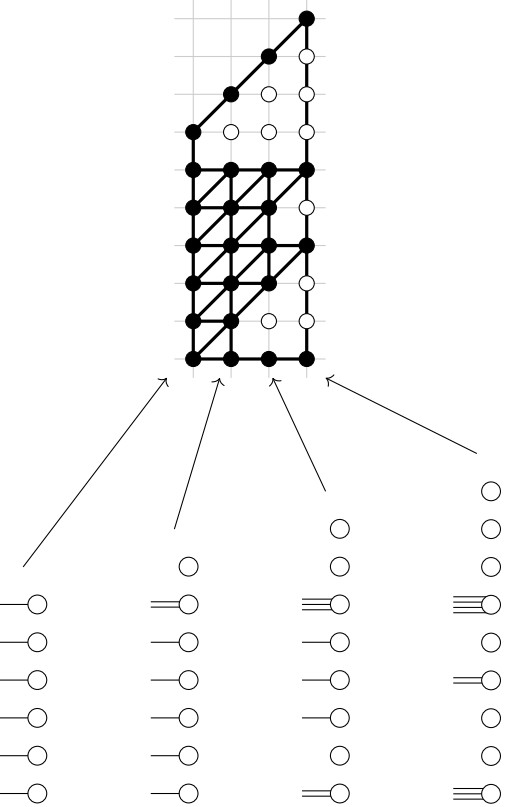

Figure 8: On top, we depict a part of the resolved GTP for the $T_9$ theory, with white dots on the right edge, along with an internal triangulation consistent with the white dots. The presence of white dots propagates into the interior of the GTP, thereby limiting the possible resolutions. Each column in the GTP corresponds to a boundary condition for D5-branes ending on D7-branes. This information is encoded in the ranks $r_i$ of the matrices appearing in $\delta T$. On the bottom, we show the associated D5-brane boundary conditions (where each circle denotes a D7-brane).

We also call $\mathcal{N}$ the set of nilpotent $n \times n$ complex matrices. The map

$$m : \mathcal{Q} \to \mathcal{N}, \tag{102}$$
$$\mathbf{q} \mapsto M = q_{n-1}\widetilde{q}_{n-1},$$

is certainly not injective, as given a nilpotent matrix $M$, there are infinitely many families $\{q_i, \widetilde{q}_i\}_{i=1,\dots,n-1}$ mapping to it. Let us define

$$r : \mathcal{Q} \to \mathbb{N}^{n-1}, \tag{103}$$
$$\mathbf{q} \mapsto (r_i = \mathrm{rank}(q_i\widetilde{q}_i))_{i=1,\dots,n-1}.$$

For a given $M \in \mathcal{N}$, the ranks of the bilinears in $q_i\widetilde{q}_i$ that map to $M$ are not fixed. In other words, $r(m^{-1}(M))$ contains more than one element. However, there is a unique element of $\mathbf{r}^{\min} \in r(m^{-1}(M))$ that minimizes the sum of the entries. If $M \in \mathcal{O}_\lambda$, then $\mathbf{r}_0$ is given by the partial sums of the transpose of $\lambda$,

$$\mathbf{r}^{\min} = \left( \sum_{j=n+1-i}^{n} \lambda_j^T \right)_{i=1,\dots,n-1}. \tag{104}$$

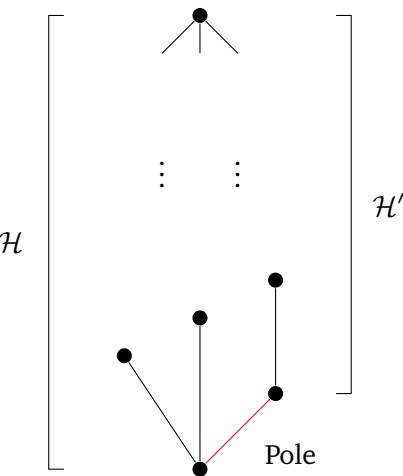

Figure 9: Schematic form of a Hasse diagram of the Higgs branch $\mathcal{H}$ viewed as a symplectic singularity. The effect of the pole prescription corresponds geometrically to imposing a higher dimensional base leaf (the transverse slice is shown in red – here it is minimal, but it does not have to be in general). The remaining Higgs branch $\mathcal{H}'$ is the transverse slice.

Note that this coincides with the ranks of the linear quiver

$$
\underset{\mathbf{r}_1^{\min}}{\bigcirc} - \underset{\mathbf{r}_2^{\min}}{\bigcirc} - \cdots - \underset{\mathbf{r}_{n-1}^{\min}}{\bigcirc} - \underset{n}{\square}
\tag{105}
$$

whose 3d $\mathcal{N} = 4$ Coulomb branch is the corresponding nilpotent orbit closure. The ranks in (104) represent the minimal deformations needed in $\delta T$ to produce the pole (99). In simple cases, the quivers (105) appear as embedded in the magnetic quiver describing the Higgs branch of the 5d theory. For a more precise statement about the embedding, the reader is referred to [55].

In the GTP, these ranks can be encoded with white dots *inside* the polygon, using again the notation introduced in [16]. For instance, for $n = 9$ and partition $\lambda = [4, 3, 2]$, we can draw the configuration shown in figure 8. The minimal ranks correspond to the number of white dots in each column: here we get $\mathbf{r}^{\min} = (0, 0, 0, 0, 0, 1, 3, 6)$. These numbers give the brane configuration which saturates the s-rule, as illustrated in the lower part of figure 8.

**Impact on the Hasse diagram.** The vevs of Higgs branch operators in the 5d theory can be projected on the space of complex structure deformations of the geometry. The fact that the pole (99) freezes some of these deformations means geometrically that the resulting pole-deformed Higgs branch is the slice in the initial Higgs branch transverse to these imposed deformations. This can be represented as follows.

The Higgs branch $\mathcal{H}$ with no pole is a symplectic singularity, which can be depicted using its Hasse diagram of singularities. The effect of the pole is to freeze certain deformations, represented here as a forced choice of a higher dimensional bottom symplectic leaf. The resulting Higgs branch $\mathcal{H}'$ is the transverse slice to that leaf. See figure 9 for a schematic depiction.

The analysis above applies to the theory in any phase. In the case of the IIA reduction on the resolved $A_{n-1}$ singularity shown in figure 6, the Higgs branch of the $T_n$ theory becomes the nilpotent cone of $\mathfrak{sl}_n$. The transverse slices are then identified with the Slodowy slices. The example $n = 4$ is illustrated in figure 10. If $M \in \mathcal{O}_\lambda$ for $\lambda$ some partition of $n$, then the corresponding Higgs branch is the Slodowy slice $\mathcal{S}_\lambda \cap \mathcal{N}$. Moving on to the SCFT phase, the Higgs branch is no longer a nilpotent orbit closure, but the general picture stays the same:

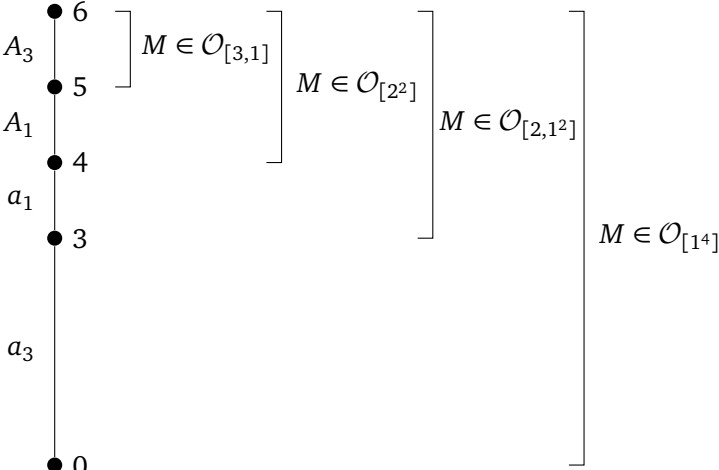

Figure 10: Diagram for the nilpotent cone of $\mathfrak{sl}_4$. The dots are nilpotent orbits. When $M$ belongs to a given orbit, the Higgs branch of the resulting theory is the transverse Slodowy slice, represented by a bracket on the right.

the Higgs branch is restricted to a transverse slice. This is what we already mentioned in the introduction, see figure 5. The Hasse diagrams drawn in figure 5 can be reproduced independently using the quiver subtraction algorithm [92, 93] on the magnetic quivers extracted from the GTPs.

### 3.6 Hanany-Witten moves

In the previous section, we defined the IIA counterpart of a 'white dot' as a nilpotent residue for the adjoint complex scalar on the flavor D6-branes. The nilpotency implies that there will be no repercussions on the geometry of the branes, and hence, the M-theory CY will not be deformed. This is the hallmark of a 'T-brane' [74].

We would now like to determine how the T-brane configuration impacts the physics of the 5d theory. This is done using the brane-web language, with Hanany-Witten moves, as we have explained in detail in section 3.3. The key point is that the nilpotent orbit of $M$ determines which Hanany-Witten move can be performed in order to detach any of the 7-branes.

In the IIA setup, flavor D6-branes are now allowed to move across exceptional $\mathbb{P}^1$s, thereby changing the quiver structure. The way this is seen at the level of the Higgs field, is that by activating vevs along the Slodowy (transverse) slice to the nilpotent vev with poles, the characteristic polynomial of the tachyon matrix actually becomes deformed.

This can be seen by computing the self-Ext group $\mathrm{Ext}^1\left(B_F^{(\mathrm{nil})}, B_F^{(\mathrm{nil})}\right)$ of the nilpotent configuration on the flavor brane. Using the computation from section 2, we are looking for $\delta T_F$ such that

$$\delta T_F \sim \delta T_F + \frac{z_2}{Z}[M, g]. \tag{106}$$

This is equivalent to requiring that $\delta T_F$ be on a transverse slice to $M$, gauge equivalent to the Slodowy slice. We will choose a gauge such that $\delta T_F$ be the companion matrix to $M$ as in [94], which is referred to as a 'reconstructible Higgs' in [74]. The details on how this is done are given in Appendix A. Say $T_{\mathrm{nil}}$ is in the maximal nilpotent orbit, then the 'Hanany-Witten'

tachyon will take the form

$$
T_{\mathrm{HW}} = T_{\mathrm{nil}} + \delta T_F = \frac{z_2}{Z}
\begin{pmatrix}
Z & 1 & & & & \\
& Z & 1 & & & \\
& & \ddots & \ddots & \ddots & \\
& & & & Z & 1 \\
(-)^{n-1}a_n X & (-)^{n-2}a_{n-1}X & & & -a_2 X & Z
\end{pmatrix},
\tag{107}
$$

where the $a_i$ are constants (the fluctuation $\delta T_F$ is proportional to $X$ as a consequence of (94) with $i = N - 1$). This matrix has determinant

$$
\det(T_{\mathrm{HW}}) = \left(\frac{z_2}{Z}\right)^n \left(Z^n + a_2 X Z^{n-2} + \ldots + a_n X\right).
\tag{108}
$$

More generally, for $M$ in the $[\lambda_1, \lambda_2, \ldots, \lambda_r]$ partition, we take a block diagonal matrix where each block will take the form:

$$
(T_{\mathrm{HW}})_i = \frac{z_2}{Z}
\begin{pmatrix}
Z & 1 & & & & \\
& Z & 1 & & & \\
& & \ddots & \ddots & \ddots & \\
& & & & Z & 1 \\
(-)^{\lambda_i-1}a_{\lambda_i}^{(i)}X & (-)^{\lambda_i-2}a_{\lambda_i-1}^{(i)}X & & & -a_2^{(i)}X & Z
\end{pmatrix}.
\tag{109}
$$

Note that by doing so, we are not using the full Slodowy slice $\mathcal{S}_\lambda$ (which contains non-block-diagonal matrices) but instead restrict to the intersection $\mathcal{S}_\lambda^0 = \mathcal{S}_\lambda \cap \mathfrak{l}_\lambda$ with the Levi subalgebra $\mathfrak{l}_\lambda$, see Appendix A. This is physically justified, as it guarantees that the flavor symmetry will not be further broken by the Hanany-Witten moves than it already has been by the white dots. The missing parameters, in $\mathcal{S}_\lambda - \mathcal{S}_\lambda^0$, are associated to the non splitting of the flavor branes, illustrated on an example below in (117).

Putting together all these blocks, and the non-flavor part of the tachyon matrix, the end result of the full deformation is

$$
Y \mapsto \frac{1}{X} \cdot \prod_{i=1}^{r} \left(Z^{\lambda_i} + X\left(\sum_{j=0}^{\lambda_i-2} a_{\lambda_i-j}^{(i)} Z^j\right)\right),
\tag{110}
$$

where $\sum_i \lambda_i = n$.

Actually it can be argued that only the coefficients $a_{\lambda_i}^{(i)}$ affect the physics near the singularity: coefficient $a_{\lambda_i-j}^{(i)}$ for $j \neq 0$ correspond to a shift of $\frac{Z^{\lambda_i-j}}{X}$, which is using (93) the coordinate along a $\mathbb{P}^1$ with which the brane has zero intersection. Therefore the equation simplifies to

$$
Y \mapsto \frac{1}{X} \cdot \prod_{i=1}^{r} \left(Z^{\lambda_i} + X a^{(i)}\right),
\tag{111}
$$

where we have renamed $a^{(i)} := a_{\lambda_i}^{(i)}$. Reverting to the original notations $(W_1, W_2, W_3, Z)$ for the coordinates in $\mathbb{C}^4$, see (52), one finally gets

$$
W_1 W_2 W_3 = \prod_{i=1}^{r} \left(Z^{\lambda_i} + a^{(i)} W_1\right).
\tag{112}
$$

Table 1: Equations defining the brane configurations in $T_3$ and $T_4$ with white dots on one edge after HW moves. In the last column, the equation is rewritten in terms of the toric variables for the resolution, and maximally factorized. The factor in orange give the ranks of the low energy quiver while the terms within brackets give the flavor ranks.

| Partition | $\det T$ | Factorization |
|-----------|----------|---------------|
| $[1^3]$ | $Y$ | $e_1 e_2^2 (z_2^3)$ |
| $[2,1]$ | $Y + a_2 Z$ | $e_1 e_2 (z_2)(e_2 z_2^2 + a_2 z_1)$ |
| $[3]$ | $Y + a_2 Z + a_3$ | smooth |

| Partition | $\det T$ | Factorization |
|-----------|----------|---------------|
| $[1^4]$ | $Y$ | $e_1 e_2^2 e_3^3 (z_2^4)$ |
| $[2,1^2]$ | $Y + a_2 Z^2$ | $e_1 e_2^2 e_3^2 (e_3 z_2^2 + a_2 z_1^2 e_1)(z_2^2)$ |
| $[2^2]$ | $Y + (a_2^{(1)} + a_2^{(2)})Z^2 + a_2^{(1)} a_2^{(2)} X$ | $e_1 e_2^2 e_3 (e_3 z_2^2 + a_2^{(1)} z_1^2 e_1)(e_3 z_2^2 + a_2^{(2)} z_1^2 e_1)$ |
| $[3,1]$ | $Y + a_2 Z^2 + a_3 Z$ | $e_1 e_2 e_3 (e_2 e_3^2 z_2^3 + a_2 z_1^2 e_1 e_2 e_3 z_2 + a_3 z_1)(z_2)$ |
| $[4]$ | $Y + a_2 Z^2 + a_3 Z + a_4$ | smooth |

**Examples.**   Let us work out a few examples. Equation (110) is worked out explicitly for $T_3$ and $T_4$ in table 1. The factorization allows to read off the corresponding quivers, which are the magnetic quivers for nilpotent orbits of $\mathfrak{sl}(3)$ and $\mathfrak{sl}(4)$ [95], as expected.

We can illustrate the problem discussed in the previous paragraph with partition $[2,1]$. The generic matrix in the Slodowy slice would give rise to the final block for the tachyon matrix given by

$$
\begin{pmatrix}
Z & 1 & 0 \\
-a_2 X & Z & \alpha X \\
\beta X & 0 & Z + \gamma X
\end{pmatrix}.
\tag{113}
$$

The resulting, deformed, equation then reads

$$
\det T = e_1 e_2 \left( a_2 \gamma e_1 z_1^3 + a_2 z_1 z_2 + \gamma e_1 e_2 z_1^2 z_2^2 + \alpha \beta e_1 z_1^3 + e_2 z_2^3 \right),
\tag{114}
$$

from which one reads off a quiver with two abelian nodes and a single flavor brane *system*:

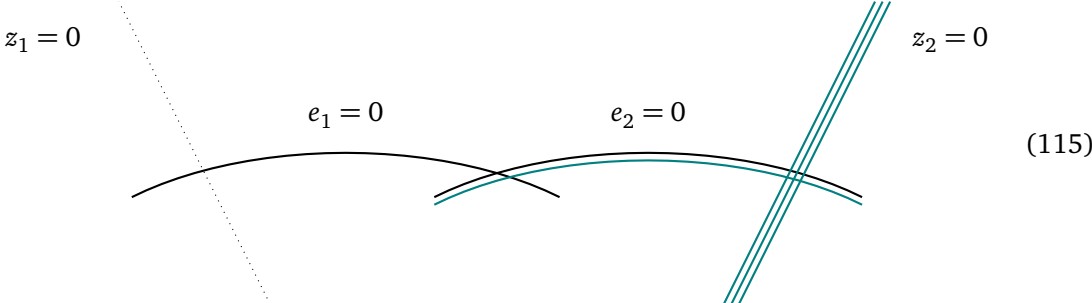

$$
\tag{115}
$$

This system, drawn in teal, intersects both $\mathbb{P}^1$ divisors at $e_1 = 0$ and $e_2 = 0$. This breaks the global isometry $\mathfrak{u}(1)$ of the nilpotent Slodowy slice $\mathcal{S}_{[2,1]} \cap \mathcal{N}$. By restricting to the Levi subalgebra, i.e. sending $\alpha$, $\beta$ and $\gamma$ to zero, one gets one more factorization:

$$
\det T = e_1 e_2 (z_2) \left( a_2 z_1 + e_2 z_2^2 \right).
\tag{116}
$$

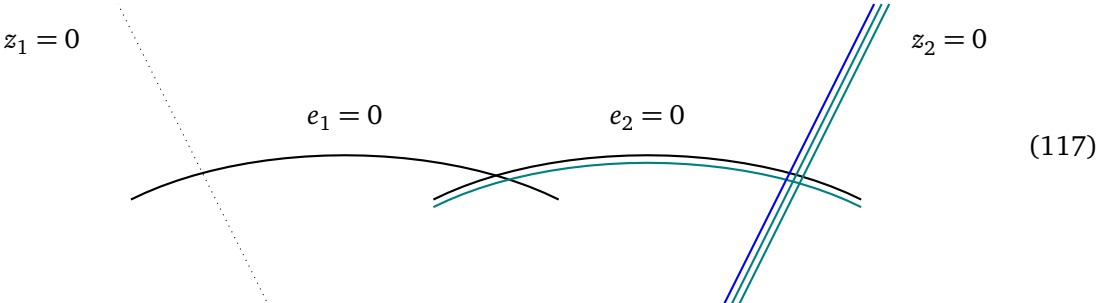

(117)

The flavor branes are now disjoint and can be moved independently:

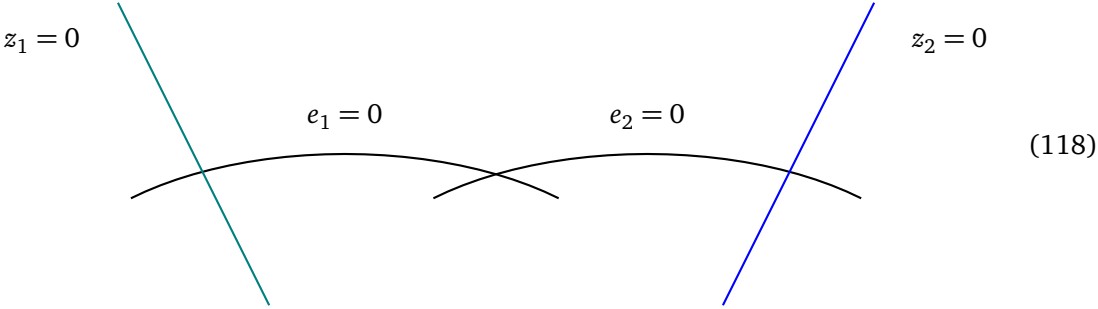

(118)

Each flavor brane intersects a single $\mathbb{P}^1$, and the global symmetry of the resulting Higgs branch (still in the resolved phase) is read to be $\mathfrak{s}(\mathfrak{u}(1) \oplus \mathfrak{u}(1)) = \mathfrak{u}(1)$.

## 4 General discussion

### 4.1 White dots along a single edge

Having discussed the case of $T_n$ in detail, we move to a generic toric polygon and its GTP deformations. Let $P$ be a convex polygon in $\mathbb{R}^2$ with vertices in $\mathbb{Z}^2$. Pick an edge on $P$, and denote by $n$ its length (i.e. the edge contains exactly $n+1$ points in $\mathbb{Z}^2$). Using an $\mathrm{SL}(2, \mathbb{Z})$ transformation, one can assume without loss of generality that this edge extends between the vertices $(0, 0)$ and $(0, n)$, and that all vertices of $P$ other than $(0, 0)$ and $(0, n)$ have coordinates $(i, j)$ with $i < 0$:

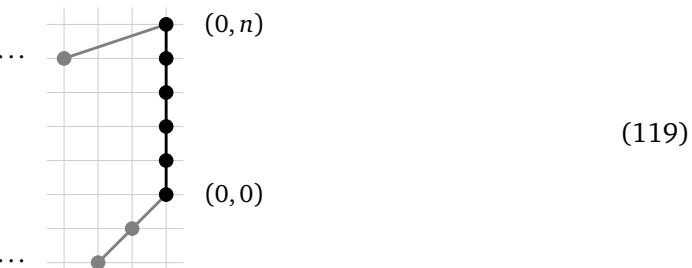

(119)

In the toric threefold, the length $n$ edge is translated into the presence of a dimension 1 singular stratum with transverse slice of type $A_{n-1}$. In the brane-web picture, this means that $n$ D5-branes extend to infinity, and the boundary condition, encoded in how they end on D7-branes, can again be encoded in a T-brane datum. This can be done explicitly using the IIA reduction as in section 3 for any given polygon (and we do so for the example of a rectangular $P$ in the next section), even though it would be extremely cumbersome to try to give general formulas that would apply to any polygon. The general idea, however, is clear: in the $T_n$ example discussed in section 3, the undeformed equation $W_1 W_2 W_3 = Z^n$ yields the $A_{n-1}$ singularity transverse to

the line $W_2 = W_3 = Z = 0$ by setting $W_1$ equal to a constant, say $W_1 = c$, giving

$$cW_2W_3 = Z^n \,. \tag{120}$$

It is then this equation that is modified by $(i)$ adding a nilpotent pole of the form (99) with $M$ in the prescribed nilpotent orbit, and $(ii)$ performing HW moves to deform the geometry, in effect replacing

$$Z^n \to \prod_i (Z^{\lambda_i} + a^{(i)} W_1) \,. \tag{121}$$

The general case is obtained by mimicking this procedure: among the equations for the toric threefold corresponding to the polygon $P$, the line with transverse singularity $A_{n-1}$ can be parametrized by a coordinate $W_1$, and the transverse slice is again given by (120). This is one of the equations defining the toric threefold, called a "boundary equation" in [96]. This equation should then be replaced by

$$cW_2W_3 = \prod_i (Z^{\lambda_i} + a^{(i)} W_1) \,. \tag{122}$$

This is in agreement with [96, Theorem 4.8]. A useful mnemonic is that in the toric diagram (41), the deformation of the edge labelled by $W_1$ involves an $A_{n-1}$ equation involving the coordinates labelling the *adjacent* edges, here $W_2$ and $W_3$, with the degree $n$ polynomial in $Z$ deformed by powers of $W_1$. In the following subsections, we give examples to illustrate the scope of our results, and also show certain limitations.

**Successive deformations.**  Before discussing the examples, we want to address a natural question: given a GTP with white dots on an edge of length $n$ characterized by a partition $\mu$, is it possible to deform it to the same GTP with partition $\lambda$ instead? One necessary condition is that $\lambda > \mu$, and it is also sufficient if the polygon is large enough.[6] In this case, one can deform the nilpotent pole according to the general rules spelled out in section 2.

For instance, if $n = 4$, we can use the explicit form (39). Going from partition $[2, 1^2]$ to $[2^2]$ is straightforward as it just involves merging two Jordan blocks, while going from $[2^2]$ to $[3, 1]$ requires using higher powers of $z$ (recall (37)):

$$
\begin{pmatrix} z & 1 & 0 & 0 \\ 0 & z & 0 & 0 \\ 0 & 0 & z & 0 \\ 0 & 0 & 0 & z \end{pmatrix}
\qquad
\begin{pmatrix} z & 1 & 0 & 0 \\ 0 & z & 0 & 0 \\ 0 & 0 & z & \alpha \\ 0 & 0 & 0 & z \end{pmatrix}
\qquad
\begin{pmatrix} z & 1 & 0 & 0 \\ 0 & z & z\beta & 0 \\ 0 & 0 & z & \alpha \\ 0 & 0 & 0 & z \end{pmatrix} \,.
$$
(123)

However it is worth mentioning that this can be done on the T-brane *before* HW deformations are added. Crucially, the two operations do *not* commute in general. Here for instance the deformed equations for $T_4$ with partition $[2, 2]$ and $[3, 1]$, given in table 1, cannot be deformed from one to the other.

---

[6]If the GTP is small, then certain deformations could lead to violations of the S-rule.

## 4.2 Rectangular box

Consider a rectangular box of size $n \times m$,

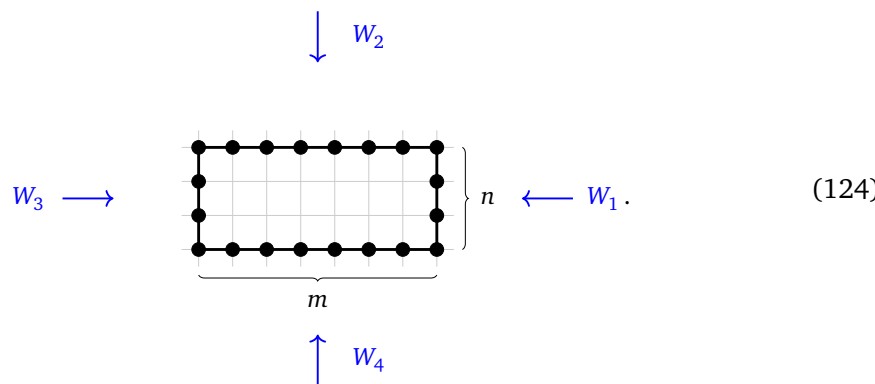

$$\tag{124}$$

The 5d SCFT this describes admits several well-known low energy gauge theory deformations. The toric threefold singularity is not a hypersurface, but it is described by the pair of equations

$$W_1 W_3 = Z^m, \qquad W_2 W_4 = Z^n. \tag{125}$$

We can add white dots on the right segment, labeled by $W_1$, exactly as in section 3. The deformed equations are

$$W_1 W_3 = Z^m, \qquad W_2 W_4 = \prod_i (Z^{\lambda_i} + a^{(i)} W_1). \tag{126}$$

Note that again, the coordinate $W_1$ is used to deform the $A_{n-1}$ equation involving the coordinates labeling the adjacent edges $W_2$ and $W_4$.

**Example.** The following example is a useful check of this proposal, as the resulting model is related to $T_3$. Consider the case $m = 3$, $n = 2$. Placing a white dot on the length 2 edge yields the equations

$$W_1 W_3 = Z^3, \qquad W_2 W_4 = Z^2 + a W_1. \tag{127}$$

We can now use the second equation to solve for $W_1$, and we get the hypersurface

$$W_2 W_3 W_4 = Z^2 (aZ + W_3). \tag{128}$$

In the limit where $a \to \infty$, the D7-brane, in the web interpretation, has crossed the whole brane system from right to left. We are left with

$$W_2 W_3 W_4 = Z^3, \tag{129}$$

and we have reproduced the prediction (5)

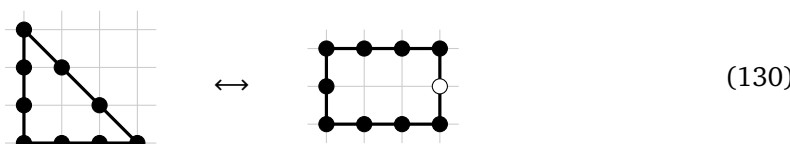

$$\tag{130}$$

This is a consistency check on the validity of our proposal.

## 4.3 Generic triangle

In this subsection we consider more examples which involve threefolds what are not hypersurfaces in $\mathbb{C}^4$. Consider the case where the toric polygon is a triangle, of arbitrary shape. Pick one edge of length $n$. Then an $\text{SL}(2,\mathbb{Z})$ transformation can bring the triangle to a frame where its three vertices are:

$$\{(0,0); (0,n); (-a,b)\}, \qquad a \in \mathbb{Z}_{>0}, \quad b \in \mathbb{Z}. \tag{131}$$

This means the toric fan is generated by the three vectors $(0,0,1)$, $(0,n,1)$ and $(-a,b,1)$. Among the generators of the dual cone we have

$$(-1,0,0) \leftrightarrow W_1, \tag{132}$$

$$\left(\frac{n-b}{d_2}, -\frac{a}{d_2}, \frac{an}{d_2}\right) \leftrightarrow W_2, \tag{133}$$

$$\left(\frac{b}{d_1}, \frac{a}{d_1}, 0\right) \leftrightarrow W_3, \tag{134}$$

$$(0,0,1) \leftrightarrow Z. \tag{135}$$

In order to write the equation of the threefold, we have introduced $d_1 = \gcd(a,b)$, $d_2 = \gcd(a, n-b)$ and $d_3 = \gcd(a, b, n-b)$. One of the equations for the toric threefold is

$$W_1^{n/d_3} W_2^{d_2/d_3} W_3^{d_1/d_3} = Z^{an/d_3}, \tag{136}$$

but in general there are other equations, as there are other generators in the dual cone. In the case of $T_n$ (41), equation (136) is sufficient and reduces to (46), but this is a special case. To illustrate this phenomenon, we consider an example.

**Example.** We take the case $a = 2$ and $b = 0$.

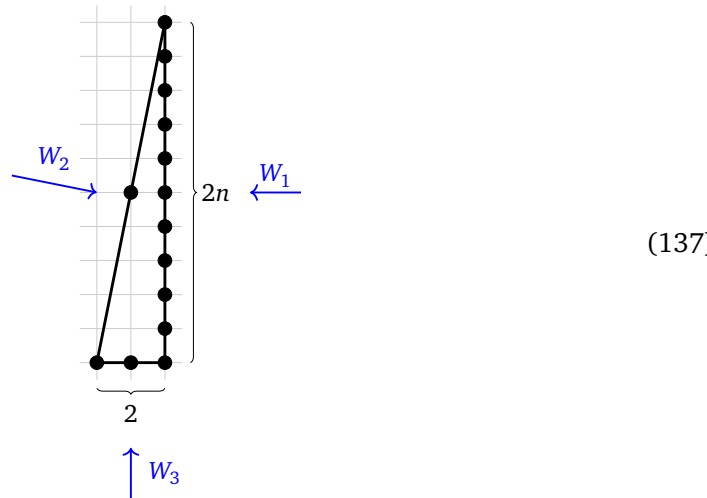

$$\tag{137}$$

The dual cone is generated by 5 vectors:[7]

$$(-1, 0, 0) \leftrightarrow W_1 \,, \tag{138}$$
$$(n, -1, 2n) \leftrightarrow W_2 \,, \tag{139}$$
$$(0, 1, 0) \leftrightarrow W_3 \,, \tag{140}$$
$$(0, 0, 1) \leftrightarrow Z \,, \tag{141}$$
$$(1, 0, 2) \leftrightarrow T \,, \tag{142}$$

and the threefold is described by two equations in $\mathbb{C}^5$:

$$W_1^n W_2 W_3 = Z^{2n} \,, \qquad W_1 T = Z^2 \,. \tag{143}$$

The first equation corresponds to (136). The singular locus associated to the length $2n$ edge is $Z = W_2 = W_3 = T = 0$, parametrized by $W_1$. Setting as before $W_1 = c$, we can eliminate $T$ and we find as expected an equation $c^n W_2 W_3 = Z^{2n}$ that we can deform. Therefore, the system of equations for the triangle (137) with one white dot on the long edge according to our conjecture is

$$\leftrightarrow \qquad \begin{cases} W_1^n W_2 W_3 = Z^{2n-2}(Z^2 + a W_1) \,, \\ W_1 T = Z^2 \,. \end{cases} \tag{144}$$

## 5 Testing the proposal: Resolution of singularities

To test the proposal for the deformation of singularities, we compute the resolutions for the deformed singularities. For the starting point, which we take to be a toric variety, the resolution is easily obtained in a combinatorial fashion, by a complete triangulation of the toric polygon. However, no such computational simplification exists yet for the resolution of GTPs. In view of this, we therefore need to revert to resolving the actual algebraic varieties. We will carry this out for the $T_n$ theories.

### 5.1 Crepant resolution of $T_n$

The simplest type of singularities are the hypersurfaces that realize the $T_n$ theories. The flavor symmetry algebra is $\mathfrak{su}(n)^3$ except for $n = 3$, where we expect $\mathfrak{e}_6$. This can be easily detected from the toric geometry by extracting the set of curves that are complete intersections between compact and non-compact (i.e. flavor) divisors. The resulting so-called combined fiber diagram (CFD) [26–28] is easily read off from the toric polygon [80].

---

[7]There is an algorithmic way to see that the fifth vector $(1, 0, 2)$ is needed, an no other. This is the computation of the Hilbert basis of the semi-group $\sigma^\vee \cap M$, using standard notation in toric geometry. This Hilbert basis is unique, and in the present case it has 5 elements, given here.

Here we will take the more laborious path of resolving the hypersurface singularity, in preparation for resolving the deformed singularities. The hypersurface in $\mathbb{C}^4$ is

$$W_1 W_2 W_3 = Z^n. \tag{145}$$

The first resolution for all of these singularities is simply to remove the locus $W_1 = W_2 = W_3 = Z = 0$, which is achieved by inserting a $\mathbb{P}^3$ with projective coordinates $[W_1, W_2, W_3, Z]$ (for a detailed exposition of resolutions from a physicist's perspective, see [97]). Denoting the exceptional section of the blowup by $\delta_1$ the equation, after proper transform, which ensures that the resolution is crepant, becomes

$$W_1 W_2 W_3 = Z^n \delta_1^{n-3}. \tag{146}$$

This can be further resolved by consecutively blowing up the loci $W_1 = W_2 = W_3 = \delta_i = 0$ i.e. by inserting another projective space with coordinates

$$[W_1, W_2, W_3, \delta_i], \qquad i = 1, \ldots, \lfloor n - 3i \rfloor, \tag{147}$$

which implies that the coordinates cannot vanish at the same time (and thus the above is a relation in the Stanley-Reissner ideal of the hypersurface). This is iterated until we reach

$$W_1 W_2 W_3 = Z^n \prod_{i=1}^{\lfloor n-3i \rfloor} \delta_i^{n-3i}. \tag{148}$$

The remaining blowups will resolve the local singularities along $W_1 = 0$, $W_2 = 0$ and $W_3 = 0$ respectively, by small resolutions, i.e.

$$[W_i, Z], \quad \text{or} \quad [W_i, \delta_i]. \tag{149}$$

Let us denote the compact divisors by $S_i$ and the non-compact divisors by $D_a$. We then find from these resolutions the intersection matrix

$$\mathcal{G}_{ab} = \sum_i S_i \cdot D_a \cdot D_b, \tag{150}$$

to be precisely the adjacency matrix of the CFD for $T_n$: i.e. three $\mathfrak{su}(n)$ Cartan matrices, pairwise connected by $-1$ curves. We have shown examples in figures 11 and 12.

## 5.2 Resolution of deformations of $T_n$

In this subsection, we will perform partial resolutions of the deformed singularities. This can be regarded as going on a non-generic subspace of the Coulomb branch that still preserves some unbroken non-Abelian flavor symmetry. Once we have blown-up a particular compact cycle, we will consider wrapped M2-branes on two-cycles within the exceptional four-cycles, and compute their charges w.r.t. the unbroken flavor symmetry. Since these cycles have non-zero Kähler volume, they are massive states.

**Deformations of $T_3$.** By including the deformations, some of the above resolutions get obstructed. Lets consider the simplest case of $T_3$, which itself is the rank 1 Seiberg theory with flavor symmetry $\mathfrak{e}_6$. After a single blowup $[W_1, W_2, W_3, Z]$ we get one compact divisor $\delta_1 = 0$, and three $A_2$ singularities. Resolving yields the CFD shown in figure 11 on the left hand side. The representations are $(\mathbf{3}, \bar{\mathbf{3}}, \mathbf{1})$, and cyclic permutations, and thus we get the known enhancement to $\mathfrak{e}_6$,[8]

$$(\mathbf{3}, \bar{\mathbf{3}}, \mathbf{1}) \oplus (\mathbf{1}, \mathbf{3}, \bar{\mathbf{3}}) \oplus (\bar{\mathbf{3}}, \mathbf{1}, \mathbf{3}) \longrightarrow \quad \mathbf{27}. \tag{151}$$

---

[8]From the geometry we are also able to extract the flavor symmetry *group* [42,98], which however will not play a role in the present paper.

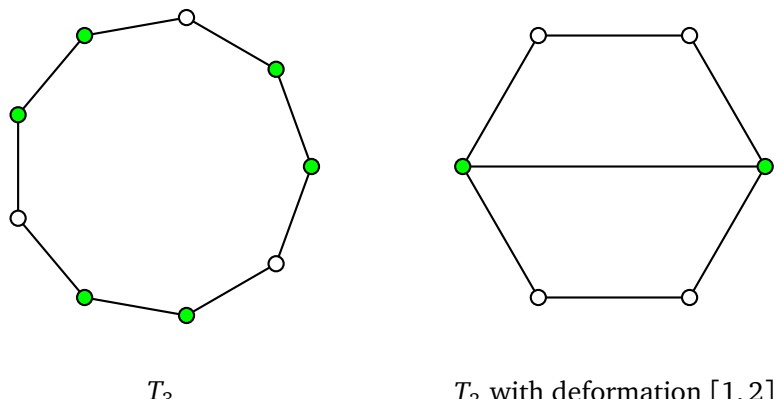

$T_3$                                         $T_3$ with deformation $[1,2]$

Figure 11: CFDs: The intersection graph for $T_3$ (left hand side) and the deformation of $T_3$ described by the partition $[1,2]$ (right). The nodes are curves that are complete intersections between $\sum S_i$ (i.e. the sum of all compact divisors) and the non-compact divisors. The green nodes are $-2$ self-intersection curves, which correspond to roots of the flavor symmetry algebra, white are $-1$ curves, which can be thought of as bifundamental matter. On the left we see the $\mathfrak{su}(3)^3$ flavor symmetry and the matter in the $(\mathbf{3}, \mathbf{3}, \mathbf{1})$ etc, which enhances to $\mathfrak{e}_6$. On the right the theory is rank 0.

Adding the deformation results in $W_1 W_2 W_3 = Z(Z^2 + W_1)$, which does not allow for a big resolution. There are small resolutions, along e.g. $W_1 = Z = 0$, indicating that this is a rank 0 theory.

**Deformations of $T_4$.** More interestingly we can start with $T_4$. All deformations involving a single edge and yielding a positive rank theory are shown in table 2.

The deformation that is associated to the partition $[2, 1^2]$ along one edge is the hypersurface

$$W_1 W_2 W_3 = Z^2 (Z^2 + W_1). \tag{152}$$

The first resolution (147) results in

$$W_1 W_2 W_3 = \delta_2 Z^4 + Z^2 W_1. \tag{153}$$

Continuing the blowup results in the intersection matrix between the sum of compact divisors $\sum_i S_i$ and each of the non-compact ones

$$\left( \sum_i S_i \right) \cdot D_a \cdot D_b = \begin{pmatrix} -1 & 1 & 0 & 0 & 0 & 0 & 0 & 0 & 1 & 0 & 0 & 0 \\ 1 & -1 & 0 & 0 & 0 & 0 & 0 & 0 & 0 & 0 & 0 & 1 \\ 0 & 0 & -2 & 1 & 0 & 0 & 0 & 0 & 0 & 1 & 0 & 0 \\ 0 & 0 & 1 & -1 & 1 & 0 & 0 & 0 & 0 & 0 & 0 & 0 \\ 0 & 0 & 0 & 1 & -2 & 0 & 0 & 0 & 1 & 0 & 0 & 0 \\ 0 & 0 & 0 & 0 & 0 & -2 & 0 & 1 & 1 & 0 & 0 & 0 \\ 0 & 0 & 0 & 0 & 0 & 0 & -2 & 1 & 0 & 0 & 0 & 1 \\ 0 & 0 & 0 & 0 & 0 & 1 & 1 & -1 & 0 & 0 & 0 & 0 \\ 1 & 0 & 0 & 0 & 1 & 1 & 0 & 0 & -2 & 0 & 0 & 0 \\ 0 & 0 & 1 & 0 & 0 & 0 & 0 & 0 & 0 & -1 & 1 & 0 \\ 0 & 0 & 0 & 0 & 0 & 0 & 0 & 0 & 0 & 1 & -2 & 1 \\ 0 & 1 & 0 & 0 & 0 & 0 & 1 & 0 & 0 & 0 & 1 & -2 \end{pmatrix}_{ab}. \tag{154}$$

This is shown in figure 12, from which the manifest flavor symmetry algebra is read off from the collection of $-2$ curves

$$\mathfrak{su}(4)^2 \oplus \mathfrak{su}(2) \oplus \mathfrak{u}(1). \tag{155}$$

The additional $\mathfrak{u}(1)$ follows from the general rule on flavor symmetry algebras from CFDs as laid out in [31]. The matter is in the representations, which combine naturally into the representations of the enhanced symmetry algebra as follows

$$(\mathbf{4}, \overline{\mathbf{4}}, \mathbf{1}) \oplus (\mathbf{6}, \mathbf{1}, \mathbf{1}) \oplus (\mathbf{1}, \mathbf{6}, \mathbf{1}) \oplus (\overline{\mathbf{4}}, \mathbf{1}, \mathbf{2}) \oplus (\mathbf{1}, \mathbf{4}, \mathbf{2}) \quad \longrightarrow \quad (\mathbf{28}, \mathbf{1}) \oplus (\mathbf{8}, \mathbf{2}), \tag{156}$$

Table 2: The partition that defines the distribution of white dots in the GTP, the rank of the 5d SCFT, the flavor symmetry algebra, as well as hypersurface equation for the deformations of $T_4$.

| Partitions | Rank | Free | Symmetry | Equation | Fig. |
|---|---|---|---|---|---|
| $[1^4]$ | 3 | 0 | $\mathfrak{su}(4)^3$ | $W_1 W_2 W_3 = Z^4$ | 12 |
| $[2,1^2]$ | 2 | 0 | $\mathfrak{su}(8) \oplus \mathfrak{su}(2)$ | $W_1 W_2 W_3 = Z^2(Z^2 + W_1)$ | 12 |
| $[2^2]$ | 1 | 0 | $\mathfrak{e}_7$ | $W_1 W_2 W_3 = (Z^2 + aW_1)(Z^2 + bW_1)$ | 12 |

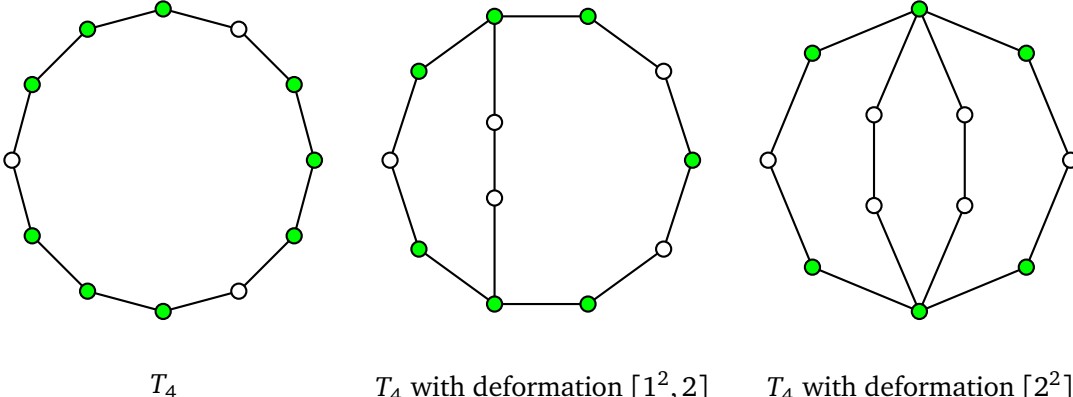

$T_4$ $\qquad\qquad$ $T_4$ with deformation $[1^2, 2]$ $\qquad$ $T_4$ with deformation $[2^2]$

Figure 12: CFDs: The intersection graph for $T_4$ (left hand side) and the deformation of $T_4$ described by the partition $[1^2 2]$ (middle) and deformation of $T_4$ with partition $[2^2]$. The nodes are curves that are complete intersections between $\sum S_i$ (i.e. the sum of all compact divisors) and the non-compact divisors. The green nodes are $-2$ self-intersection curves, which correspond to roots of the flavor symmetry algebra, white are $-1$ curves, which can be thought of as bifundamental matter. On the left we see the $\mathfrak{su}(4)^3$ flavor symmetry and the matter in the $(\mathbf{4}, \mathbf{4}, \mathbf{1})$ etc representations. In the middle, we see the manifest flavor symmetry $\mathfrak{su}(4)^2 \oplus \mathfrak{su}(2) \oplus \mathfrak{u}(1)$, but the matter shown results in the enhancement to $\mathfrak{su}(8) \oplus \mathfrak{u}(1)$. On the right the flavor symmetry enhances to $\mathfrak{e}_7$.

where we identify the enhanced flavor symmetry algebra as

$$\mathfrak{g}_{UV} = \mathfrak{su}(8) \oplus \mathfrak{su}(2) \,. \tag{157}$$

Similarly for the partition $[2^2]$, the manifest symmetry is $\mathfrak{su}(4)^2 \oplus \mathfrak{su}(2)$, while the actual symmetry is $\mathfrak{e}_7$. In the CFD, see figure 12 on the right hand side, only $\mathfrak{su}(4)^2$ appears explicitly (the $\mathfrak{su}(2)$ term is reflected in the presence of an additional $\mathbb{Z}_2$ symmetry of the diagram), and we see the $\mathfrak{su}(4)^2$ representation

$$(\mathbf{4}, \overline{\mathbf{4}}) \oplus (\overline{\mathbf{4}}, \mathbf{4}) \oplus 2 \times (\mathbf{6}, \mathbf{1}) \oplus 2 \times (\mathbf{1}, \mathbf{6}) \quad \longrightarrow \quad \mathbf{56} \,. \tag{158}$$

These indeed arise from the branching of the $\mathbf{56}$ of $\mathfrak{e}_7$ to $\mathfrak{su}(4)^2$.

**Deformations of $T_5$.** Finally consider the $T_5$ model with deformations added along a single edge. These are summarized in table 3.

First we recompute the resolution of the undeformed $T_5$ model, which is given in figure 13. The flavor symmetry algebra and the representation of the hypers is

$$\mathfrak{g}_{UV} = \mathfrak{su}(5)^3 : \qquad (\mathbf{5}, \overline{\mathbf{5}}, \mathbf{1}) \oplus (\mathbf{1}, \mathbf{5}, \overline{\mathbf{5}}) \oplus (\overline{\mathbf{5}}, \mathbf{1}, \mathbf{5}) \,. \tag{159}$$

Table 3: Deformations of the $T_5$ model: the partition that defines the GTP, the rank of the expected 5d SCFT, the flavor symmetry algebra, and the deformed hypersurface equation.

| Partitions | Rank | Symmetry | Equation | Fig. |
|:---:|:---:|:---:|:---:|:---:|
| $[1^5]$ | 6 | $\mathfrak{su}(5)^3$ | $W_1 W_2 W_3 = Z^5$ | 13 |
| $[2, 1^3]$ | 5 | $\mathfrak{su}(5)^2 \oplus \mathfrak{su}(3) \oplus \mathfrak{u}(1)$ | $W_1 W_2 W_3 = Z^3(Z^2 + W_1)$ | 13 |
| $[2^2, 1]$ | 4 | $\mathfrak{su}(5)^2 \oplus \mathfrak{su}(2) \oplus \mathfrak{u}(1)$ | $W_1 W_2 W_3 = Z(Z^2 + W_1)^2$ | 15 |
| $[3, 1^2]$ | 3 | $\mathfrak{su}(10) \oplus \mathfrak{su}(2)$ | $W_1 W_2 W_3 = Z^2(Z^3 + W_1)$ | 13 |
| $[3, 2]$ | 2 | $\mathfrak{su}(10)$ | $W_1 W_2 W_3 = (Z^2 + W_1)(Z^3 + W_1)$ | 14 |

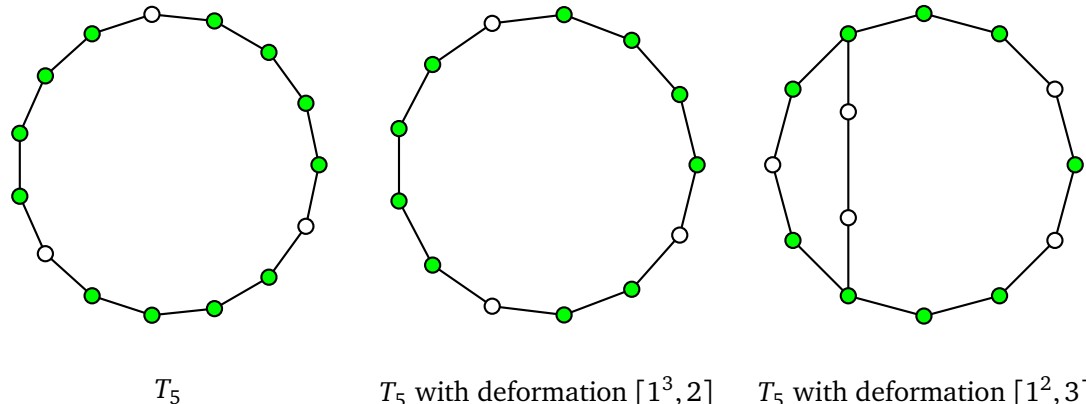

$T_5$        $T_5$ with deformation $[1^3, 2]$     $T_5$ with deformation $[1^2, 3]$

Figure 13: CFDs: The intersection graph of $-2$ (green) and $(-1)$ (white) curves, i.e. the CFD, for $T_5$ (LHS), the partition $[2, 1^3]$ in the middle, and $[3, 1^2]$ on the right hand side.

The CFD is shown in figure 13. Adding a single white dot results in a theory with flavor algebra $\mathfrak{su}(5)^2 \oplus \mathfrak{su}(3) \oplus \mathfrak{u}(1)$, with the CFD shown in the middle in figure 13, which has bifundamental matter consistent with these flavor symmetry algebras.

Adding two white dots along a single edge in the partition $[3, 1^2]$ we obtain the right hand figure 13. The representations under $\mathfrak{su}(5)^2 \oplus \mathfrak{su}(2)$ are

$$(\mathbf{5}, \overline{\mathbf{5}}, \mathbf{1}) \oplus (\mathbf{1}, \mathbf{5}, \mathbf{2}) \oplus (\overline{\mathbf{5}}, \mathbf{1}, \mathbf{2}) \oplus (\mathbf{10}, \mathbf{1}, \mathbf{1}) \oplus (\mathbf{1}, \overline{\mathbf{10}}, \mathbf{1}) \quad \longrightarrow \quad (\mathbf{45}, \mathbf{1}) \oplus (\overline{\mathbf{10}}, \mathbf{2}), \tag{160}$$

which is consistent with the flavor symmetry enhancement to

$$\mathfrak{g}_{UV} = \mathfrak{su}(10) \oplus \mathfrak{su}(2). \tag{161}$$

Similarly for $[3, 2]$ we computed the resolution and find the CFD shown in figure 14, which shows the representations under $\mathfrak{su}(5)^2$ to be

$$(\mathbf{5}, \overline{\mathbf{5}}) \oplus (\mathbf{10}, \mathbf{1}) \oplus (\mathbf{1}, \overline{\mathbf{10}}) \quad \longrightarrow \quad \mathbf{45}, \tag{162}$$

which is consistent with the enhancement to $\mathfrak{su}(10)$. Finally consider figure 15, where we manifestly see the $\mathfrak{su}(5)$ but the $\mathfrak{su}(2)$ is decomposed into $\mathfrak{u}(1)$.

For partition $[2^2, 1]$, although the $\mathfrak{su}(2)$ factor of the global symmetry is not directly visible in the resolution, an educated guess allows to read the following representations of $\mathfrak{su}(5) \oplus \mathfrak{su}(5) \oplus \mathfrak{su}(2)$ on figure 15:

$$(\mathbf{5}, \mathbf{1}, \mathbf{1}) \oplus (\mathbf{1}, \overline{\mathbf{5}}, \mathbf{1}) \oplus (\mathbf{10}, \mathbf{1}, \mathbf{2}) \oplus (\mathbf{5}, \overline{\mathbf{10}}, \mathbf{2}) \oplus (\overline{\mathbf{5}}, \mathbf{5}, \mathbf{1}). \tag{163}$$

Again, the flavor symmetry algebra is consistent with the one predicted from GTPs.

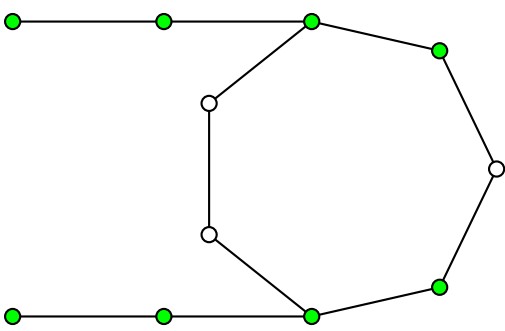

$T_5$ with deformation [2,3]

Figure 14: CFDs: The intersection graph for $T_5$ with the deformation labeled by the partition $[2,3]$. We see the $\mathfrak{su}(5)^2 \oplus \mathfrak{u}(1)$ and the matter in the $(\mathbf{5},\mathbf{5}) \oplus (\mathbf{10},\mathbf{1}) \oplus (\mathbf{1},\mathbf{10})$ which enhance into $\mathbf{45}$ of $\mathfrak{su}(10)$.

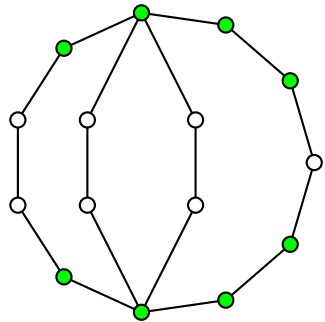

$T_5$ with deformation $[1,2^2]$

Figure 15: CFDs: The intersection graph for $T_5$ with the deformation labeled by the partition $[1,2^2]$.

# Acknowledgments

We would like to thank Cyril Closset for collaboration in early stages of this work. AB and SSN thank Hendrik Süß for discussions of related questions.

**Funding information**   This work, in particular AC and SSN, was supported and enabled by the Fondation Wiener-Anspach. This research is further supported by IISN-Belgium (convention 4.4503.15). AB is supported by the ERC Consolidator Grant 772408-Stringlandscape, and by the LabEx ENS-ICFP: ANR-10-LABX-0010/ANR-10-IDEX-0001-02 PSL*. SSN is supported in part by the "Simons Collaboration on Special Holonomy in Geometry, Analysis and Physics" and the EPSRC Open Fellowship EP/X01276X/1.

# A   Basic algebraic notions for $\mathfrak{sl}_n$

This appendix gathers a few elementary algebraic concepts needed in the bulk of the paper. In the following, we take $\mathfrak{g} = \mathfrak{sl}_n(\mathbb{C})$, which we represent as $n \times n$ complex matrices with vanishing trace, and we pick the diagonal matrices for the Cartan subalgebra $\mathfrak{h}$.

**Nilpotent orbits and triples.** Let $e$ be a nilpotent element of $\mathfrak{g}$. By the Jacobson-Morozov theorem, one can construct an $\mathfrak{sl}_2$ triple $(e, f, h)$, i.e. a triple of elements of $\mathfrak{g}$ with $h \in \mathfrak{h}$ satisfying the commutation relations

$$[e, f] = h, \qquad [h, e] = 2e, \qquad [h, f] = -2f. \tag{A.1}$$

More precisely, there exists an embedding

$$\rho : \mathfrak{sl}_2 \longrightarrow \mathfrak{g}, \tag{A.2}$$

such that $\rho(e)$ defines a nilpotent element of $\mathfrak{g}$. In our case, nilpotent orbits are characterized by partitions of $n$. For the partition $\lambda = (\lambda_1, \ldots, \lambda_r)$, there is a canonical triple, where $e$ is in the Jordan form specified by $\lambda$, $h$ is diagonal and $f$ is lower diagonal, defined as follows. Consider first the maximal orbit, $\lambda = (n)$. In this case we define the canonical triple $e = J_n$, $h = H_n$ and $f = \tilde{J}_n$ where

$$(J_n)_{i,j} = \delta_{i+1,j}, \qquad (H_n)_{i,j} = \delta_{i,j}(n + 1 - 2i), \qquad (\tilde{J}_n)_{i,j} = \delta_{i-1,j}j(n - j). \tag{A.3}$$

Then for any partition $\lambda$ the canonical triple is the block diagonal

$$e = \mathrm{Diag}(J_{\lambda_i}), \qquad f = \mathrm{Diag}(\tilde{J}_{\lambda_i}), \qquad h = \mathrm{Diag}(H_{\lambda_i}). \tag{A.4}$$

**Slodowy Slices.** Given a triple $(e, f, h)$, one constructs the space

$$\mathcal{S}_e = e + \mathfrak{g}^f, \tag{A.5}$$

where $\mathfrak{g}^f$ is the centralizer of $f$ in $\mathfrak{g}$. This is called the *Slodowy slice* transverse to $e$. Note that this should not be confused with the *nilpotent Slodowy slice*, which is the intersection of the Slodowy slice with the nilpotent cone. When $(e, f, h)$ is the canonical nilpotent element (A.4) associated to partition $\lambda$, we call $\mathcal{S}_\lambda$ the Slodowy slice.

In order to construct explicitly $\mathcal{S}_\lambda$, we first need the set of matrices which commute with $\tilde{J}_n$. These are the matrices $S(a_1, \ldots, a_n)$ with $a_1, \ldots, a_n \in \mathbb{C}$, defined by[9]

$$(S(a_1, \ldots, a_n))_{ij} := \sum_{k=0}^{n-1} a_{k+1} \delta_{i-k,j} \prod_{l=j}^{i-1} l(n - l). \tag{A.7}$$

Then the centralizer $\mathfrak{g}^f$ is a set of block matrices of sizes $\lambda_i \times \lambda_j$, where the block $(i, j)$ depends on $\min(\lambda_i, \lambda_j)$ parameters. We will not need its explicit form here. We only need the form of the diagonal blocks $(i = j)$, which are of the form

$$S(a_1^{(i)}, \ldots, a_{\lambda_i}^{(i)}). \tag{A.8}$$

Using this, we can compute the dimension of the Slodowy slices, which is

$$\dim \mathcal{S}_\lambda = -1 + \sum_{1 \le i, j, \le r} \min(\lambda_i, \lambda_j) = -(n + 1) + 2 \sum_{i=1}^{r} i \lambda_i, \tag{A.9}$$

---

[9]The matrices $S(a_1, \ldots, a_n)$ for low values of $n$ are:

$$S(a_1, a_2) = \begin{pmatrix} a_1 & 0 \\ a_2 & a_1 \end{pmatrix}, \quad S(a_1, a_2, a_3) = \begin{pmatrix} a_1 & 0 & 0 \\ 2a_2 & a_1 & 0 \\ 4a_3 & 2a_2 & a_1 \end{pmatrix}, \quad S(a_1, a_2, a_3, a_4) = \begin{pmatrix} a_1 & 0 & 0 & 0 \\ 3a_2 & a_1 & 0 & 0 \\ 12a_3 & 4a_2 & a_1 & 0 \\ 36a_4 & 12a_3 & 3a_2 & a_1 \end{pmatrix}. \tag{A.6}$$

and

$$\dim \mathcal{S}_\lambda \cap \mathcal{N} = \dim \mathcal{S}_\lambda - (n-1) = -2n + 2\sum_{i=1}^{r} i\lambda_i. \tag{A.10}$$

The dimension of the nilpotent cone $\mathcal{N}$ is $n(n-1)$, and therefore we can deduce that the dimension of the nilpotent orbit of $e$ is

$$\dim \mathcal{O}_\lambda = n(n+1) - 2\sum_{i=1}^{r} i\lambda_i, \tag{A.11}$$

which matches known results.

**Levi subalgebras and Slodowy slices.** A Borel subalgebra $\mathfrak{b}$ of $\mathfrak{g}$ is a maximal solvable subalgebra. The standard choice, which we make, is to pick for $\mathfrak{b}$ the upper triangular matrices. The simple roots $\alpha_i$ can be indexed by $i \in I := \{1, \ldots, n-1\}$, and to every subset $\Theta$ of $I$ one can associate a parabolic subalgebra $\mathfrak{p}_\Theta$ of $\mathfrak{g}$ containing $\mathfrak{b}$. This parabolic subalgebra decomposes as a direct sum

$$\mathfrak{p}_\Theta = \mathfrak{l}_\Theta \oplus \mathfrak{n}_\Theta, \tag{A.12}$$

where $\mathfrak{l}_\Theta$ is the Levi subalgebra and $\mathfrak{n}_\Theta$ is the nilradical of $\mathfrak{p}_\Theta$.

Let $\lambda = (\lambda_1, \ldots, \lambda_r)$ be a partition of $n$. We associate to this partition the set

$$\Theta_\lambda = I - \{\lambda_1, \lambda_1 + \lambda_2, \ldots, \lambda_1 + \ldots + \lambda_{r-1}\}. \tag{A.13}$$

This way, one can construct the corresponding Levi subalgebra $\mathfrak{l}_\lambda := \mathfrak{l}_{\Theta_\lambda}$. We then introduce the intersection

$$\mathcal{S}_\lambda^0 := \mathcal{S}_\lambda \cap \mathfrak{l}_\lambda. \tag{A.14}$$

Without choices, $\mathfrak{l}_\lambda$ is simply the algebra of block diagonal matrices, with blocks of respective sizes $\lambda_1, \ldots, \lambda_r$, and $\mathcal{S}_\lambda^0$ is the set of traceless matrices which are block diagonal, with diagonal blocks $J_{\lambda_i} + S(a_1, a_2, \ldots, a_{\lambda_i})$. The dimension is $\dim \mathcal{S}_\lambda^0 = n - 1$, independent of $\lambda$.

**Companion matrix.** It is easy to check that the matrix $J_n + S(0, a_2, \ldots, a_n)$ can be conjugated to a matrix of the form

$$C(b_2, \ldots, b_n) := \begin{pmatrix} 0 & & 1 & & & \\ & 0 & & 1 & & \\ & & & \ddots & \ddots & \ddots & \\ & & & & & 0 & 1 \\ (-)^{n-1}b_n & (-)^{n-2}b_{n-1} & & \cdots & & -b_2 & 0 \end{pmatrix}, \tag{A.15}$$

where the $b_i$ are known functions of the $a_i$. Note that the change of basis depends explicitly on the $a_i$. A matrix in the form (A.15) is called a *companion matrix*. It has the convenient property that its characteristic polynomial (evaluated at $-z$) is

$$\det(z\mathbf{1}_n + C) = z^n + \sum_{i=2}^{n} b_i z^{n-i}. \tag{A.16}$$

Any matrix in $\mathcal{S}_\lambda^0$ can then be conjugated to a block diagonal matrix with blocks of the form $C(b_2, \ldots, b_{\lambda_i})$.

**Cokernel of ad($e$).**    Consider a finite dimensional representation $\rho : \mathfrak{sl}(2) \to V$ of $\mathfrak{sl}(2)$. One can decompose $V$ into irreducible representations $V_i$ of dimensions $n_i$. In each irrep, the image of $\rho(e)$ has codimension 1, and the cokernel is spanned by the weight space of lowest weight, which by definition is precisely the kernel of $\rho(f)$. Therefore in the Lie algebra $\mathfrak{g}$ seen as an $\mathfrak{sl}(2)$ representation defined by a triple $(e, f, h)$, the cokernel of ad($e$) is $\mathfrak{g}^f$.

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
