# Peer review of "Generalized Toric Polygons, T-branes, and 5d SCFTs"

_SciPost Physics, doi:SciPost Phys. 18, 079 (2025)_

## Round 1 · Referee Report · Anonymous (Referee 1) · 2023-3-22

Report

The paper studies constructions of 5d superconformal field theories (SCFTs) via brane webs in IIB string theory and via singular Calabi-Yau threefolds in M-theory. One of the unsolved questions has been to identify the M-theory Calabi-Yau threefold corresponding to 5-branes webs with 7-branes, where there are multiple 5-branes ending on a single 7-branes. The presence of these 7-branes makes the dual M-theory Calabi-Yau threefold non-toric. Therefore the standard duality between webs and toric polygons does not apply. The authors address this question in a restricted but still large class of examples, highlighting also the general procedure. In particular, the paper explicitly identifies the Calabi-Yau geometries corresponding to the dual generalized toric polygons (GPT). GPTs were previously introduced as dual diagram of the certain IIB brane webs with 7-branes, but they did not have a full geometric description in M-theory. Key for the understanding of the M-theory geometry is the intermediate step of the dualization procedure between IIB and M-theory. This is the IIA construction, which is given by D6-branes intersecting along genus-0 curves and forming a 5d linear quiver theory. The paper shows that, when in the web multiple 5-branes end on a 7-brane, a nilpotent T-brane background for the D6-brane configuration in IIA has been activated. Finally semisimple deformations of the T-brane configuration, which are dual to Hanany-Witten transitions in the IIB web, lead to complex structure deformations of the Calabi-Yau geometry upon uplift to M-theory. This last step is key for having an explicit geometry in M-theory. The authors finally propose some checks of this proposal, and they apply them to some explicit examples. In particular, the main test consists in resolving the resulting singular Calabi-Yau geometry, and in matching properties derived from these resolutions with the ones computed from the dual brane webs. These include flavor symmetries and/or charges of BPS states. The examples studied in this paper successfully pass the proposed tests.

The paper provide interesting new results and methodologies to address the question mentioned above, therefore I recommend it for publication to SciPost, when the following minor points about section 2 are addressed, answered or justified:

The idea of having a general section about the technical T-brane procedure is very useful. On the other hand sometimes the used terminology can be a bit too technical in the mathematical sense, and it can sound a bit cryptic to some physics readers. The general procedure does become clearer when looking at the explicit examples though.

  • To be more concrete the sentence after (2.7) " ... are monic polynomials ...". This sentence together with footnote 2 sounds a bit obscure. It might be more helpful to just show the general intuitive form of the SNF(T) matrix, or expand the discussion making it more explicit or manifest.

  • in (2.10) the partition notation on the T matrices has been introduced without explanation (only mentioned in introduction). Again, this becomes clear later on, but when reading this first, one might get confused on what this notation might mean here.

  • I do not understand how (2.14) - (2.17) reveals the structure of the deformation (2.11), in a different guise, or what "reveals the same structure in a different guise" refers to.

---

## Round 1 · Referee Report · Anonymous (Referee 2) · 2023-4-10

Strengths

  1. First proposal of geometric description of 5d SCFTs described by 5-brane webs with multiple 5-branes ending on the same 7-brane.
  2. Passes consistency checks using known deformations.

Weaknesses

  1. The Type IIA description of the deformation as 6-brane recombination is not sufficiently explained.

Report

The authors propose an M-theoretic geometric description of 5d SCFT's corresponding to 5-brane webs in Type IIB string theory
in which groups of multiple external 5-branes are constrained to end on common 7-branes.
The geometry in this case is not a toric Calbi-Yau, and the corresponding grid diagram is referred to as a Generalized Toric Polygon (GTP).
More specifically they consider a subclass of such models, in which the constraints apply to just one edge of the polygon.
Their proposal involves a deformation of the toric CY which is motivated by an intermediate description of the system in Type IIA string theory,
that involves a combination of singular geometry and 6-branes wrapping collapsed 2-cycles.
There are two steps leading to their proposal.
In the first step, they argue that the GTP itself, namely the constraints imposed on some of the external 5-branes,
corresponds in the Type IIA picture to nilpotent VEVs for the Higgs field on the 6-brane worldvolume.
Such VEVs do not separate the 6-branes, and therefore do not correspond to a modification of the geometry in the M-theory picture.
The second step is the deformation itself.
They propose an algebraic expression for the deformation, and argue that from the 5-brane-web viewpoint it corresponds
to a Hanany-Witten type move that releases the 7-branes that have multiple 5-branes ending on them.
The intermediate Type IIA picture of this step, which is what is supposed to provide the main motivation for this proposal,
is that some of the 6-branes combine and move away from the singularity, as they say below equation (3.45).
I did not really understand this process.
Is there an intuitive explanation?

Their proposal does pass consistency checks using some known deformations of the T_N theories.
I therefore believe it is correct, but I think a simple explanation of the 6-brane recombination process should be possible.

Two minor comments about section 5.2 are that
a. It would be helpful to clarify that the "matter" they discuss
corresponds to massive (and non-gauge invariant) BPS states on the Coulomb branch, since one might
mistakingly assume that the discussion is about (gauge-invariant) BPS operators of the 5d SCFT.
b. In equation (5.14) one should probably also include the SU(2) charges, i.e. the last four states are really two SU(2) doublets.

There is also a small typo in equation (4.9): In the second equation the power of Z should be 2.

Requested changes

  1. A simple explanation of the 6-brane recombination.
  2. Minor clarifications in section 5.2 (see report).

---

## Round 1 · Referee Report · Anonymous (Referee 3) · 2023-4-18

Report

In this paper, the authors take valuable steps towards generalizing the correspondence between 5D SCFTs and toric polygons to accommodate so-called generalized toric polygons (GTPs), which are further characterized by a choice of color for each node associated to specific choices of fivebrane boundary conditions. The key missing ingredient in this more general correspondence is the geometric interpretation of GTPs as local (but not necessarily toric) CY3s defining M-theory backgrounds.

The authors make a compelling proposal on how to supply this missing ingredient by interpreting the local geometries associated to a certain class of GTPs as comprising local CY3 patches modified by algebraic (i.e., complex structure) deformations associated to T-branes in the type IIA duality frame. This proposal, supported by ample technical arguments and numerous illustrative examples, appears to initiate a fruitful line of investigation with clearly defined future directions. The overarching efforts, if eventually successful, will clearly constitute significant progress towards understanding geometric engineering of 5D SCFTs in string theory.

One aspect that I find somewhat unclear is to what extent the capacity for toric polygons to be able to describe fivebranes ending on mutually non-local sevenbrane configurations is essential for the central result of this paper, especially since the authors confine their attention to mutually local configurations of sevenbranes (specifically, parallel stacks). Said differently, one might wonder whether or not the local geometric interpretation could have been spelled out simply using stacks of parallel sevenbranes, without reference to complete toric polygons, despite the fact that toric polygons nicely contextualize the results and clearly motivate future research.

My opinion is that this paper could perhaps benefit from an additional comment clarifying and/or emphasizing the extent to which the complete (generalized) toric polygons were necessary for obtaining the advertised results. With this minor revision (and leaving the specifics to the authors' discretion), I would enthusiastically recommend this paper for publication.

---

## Round 2 · Referee Report · Anonymous (Referee 1) · 2025-1-29

Report

The authors have clearly addressed all the raised minor points in the reports. Therefore I recommend the paper for publication in SciPost.

Recommendation

Publish (easily meets expectations and criteria for this Journal; among top 50%)

---

## Round 2 · Author Response

We thank the three referees for their thorough reading of our draft. We would also like to apologize for the very long delay in submitting this updated version. We were waiting for the appearance of an upcoming article in mathematics checking our results from a completely different route. This paper appeared a few months ago [2410.04714] and we have checked that their results are fully consistent with ours (as explicitly mentioned e.g. below their equation (6.9)).

---

## Round 2 · List of Changes

We have addressed the various points raised by the three referees as detailed below.

Referee 1 :
\begin{itemize}
\item We have added a more detailed explanation of the brane recombination process in section 1, around equations (1.3) to (1.9). We have also included a reference to this after equation (3.45).
\item Regarding question (a.), we have added a paragraph at the beginning of section 5.2 to clarify the origin of the matter states which are used to analyze the global symmetries.
\item Regarding question (b.) about equation (5.14), we have added an explanation about the $su(2)$ factor, whose representations can not be extracted directly from the CFD, hence the focus on the $su(4)^2$ representations only.
\item We have corrected the power of $Z$ in equation (4.9).
\end{itemize}

Referee 2 :
\begin{itemize}
\item We have added sentences below (2.7) to clarify the physical interest of the SNF.
\item The partition notation has been explicitly introduced in a paragraph on page 9. The same notation is then used in equations (2.10) and (2.27).
\item To address the requests on the Smith Normal Forms, we have slightly expanded the discussion in the Example on page 9, and we have added a clarifying sentence \textit{``This gives an algorithmic way to decide whether two tachyon maps encode equivalent physical systems. Several examples are given below. "}. We have also corrected a sign error in equation (2.11).
\end{itemize}

Referee 3 :
\begin{itemize}
\item The minor revision is addressed by adding to the introduction a sentence clarifying why the language of GTPs is used: \textit{The results of this paper could in principle be phrased purely in the language of stacks of D5 branes ending on D7 branes, dualized to a D6 setup. However, the GTP formalism is adopted as it paves the way towards the necessary generalizations to encompass arbitrary 5d SCFTs. }
\end{itemize}

---

## Editorial Decision

published